# Chromosome-scale genome assembly of the European common cuttlefish *Sepia officinalis*

**Simone Daniela Rencken[1,2], Georgi Tushev[1], David Hain[1,3], Elena Ciirdaeva[1], Oleg Simakov[4], Gilles Laurent[1]\***

[1]Max Planck Institute for Brain Research, Frankfurt am Main, Germany; [2]Radboud University, Donders Institute for Brain, Cognition and Behaviour, Nijmegen, Netherlands; [3]Faculty of Biological Sciences, Goethe University, Frankfurt am Main, Germany; [4]Department of Neuroscience and Developmental Biology, University of Vienna, Vienna, Austria

**\*For correspondence:**
g.laurent@brain.mpg.de

**Competing interest:** The authors declare that no competing interests exist.

## eLife Assessment

This manuscript reports a high-quality genome assembly of the European cuttlefish, *Sepia officinalis*, a representative species of the Cephalopod lineage. This **solid** work relies on current best practices in genome sequencing and assembly, combining PacBio HiFi long reads and Hi-C chromatin conformation capture, and on state-of-the-art comparative genomic analyses, including chromosome number evolution and analyses of expanded gene families. The resulting genome will be a **valuable** resource for researchers interested in cuttlefish biology and comparative genomics in general.

**Abstract** Coleoid cephalopods, a subclass of mollusks that includes octopuses, cuttlefish, and squid, exhibit sophisticated biological features, such as dynamic and neurally driven camouflage behavior, inter-individual communication, single-lens camera-like eyes, the largest brains among invertebrates, and a distinctive embryonic development. The common cuttlefish *Sepia officinalis* has served as a model organism in various research fields, spanning biophysics, neurobiology, behavior, evolution, ecology, and biomechanics. More recently, it has become a model to investigate the neural mechanisms underlying cephalopod camouflage, using quantitative behavioral approaches alongside molecular techniques to characterize the identity, evolution, and development of neuronal cell types. Despite significant interest in this animal, a high-quality, annotated genome of this species is still lacking. To address this, we sequenced and assembled a chromosome-scale genome for *S. officinalis*. Our assembly spans 5.68 billion base pairs and comprises 1n=47 repeat-rich chromosome scaffolds. This was unexpected because the haploid karyotypes of other decapods indicate 46 chromosomes. Detailed comparisons of our data to those from published decapod genome assemblies and to another recent genome assembly of *S. officinalis* (itself suggesting 1n=49 chromosomes) in fact revealed clear homologies between 46 scaffolds across all the datasets. In-depth comparison of datasets reveals highly repetitive regions at discordant scaffold boundaries and suggests that the true karyotype of *S. officinalis* is probably 1n=46 chromosomes, a likely ancestral and if true, conserved decapod karyotype. Our results include a comprehensive gene annotation and full-length transcript prediction, which we used to characterize orthologous gene families across mollusks. We identified several large-scale expansions specific to cephalopods, with many genes specific to neural or non-neural tissues of adult *S. officinalis*. In summary, this genome should provide a valuable resource for future research on the evolution, brain organization, information processing, development, and behavior in this important clade.

## Introduction

Coleoid cephalopods (octopus, squid, cuttlefish) are a highly derived group of mollusks, characterized by the largest nervous systems among all invertebrates (ca. 500 million neurons in adult octopus of which 200 million are in the central brain *Giuditta et al., 1971*; *Young, 1963a*, compared to ca. 140,000 in the fruit fly *Dorkenwald et al., 2024* or 70 million in the mouse *Herculano-Houzel et al., 2006*) and specializations with a great historical importance for neuroscience (e.g. 'giant axons' *Hodgkin et al., 1952* and 'giant synapses' *Llinás et al., 1981a*; *Llinás et al., 1981b*; *Llinas, 1984*). These animals exhibit very sophisticated behaviors, such as dynamic camouflage (*Hanlon and Messenger, 2018*; *Josef et al., 2016*; *Osorio et al., 2022*; *Marshall and Messenger, 1996*; *Zylinski et al., 2009a*; *Zylinski et al., 2012*; *Zylinski et al., 2009b*), learning (*Jozet-Alves et al., 2013*; *Schnell et al., 2021*; *Boycott and Young, 1955*), social communication (*Mather, 2006*; *Hall and Hanlon, 2002*; *Norman et al., 1999*) and hunting (*Sampaio et al., 2021*; *Sampaio et al., 2024*; *Feord et al., 2020*) as well as two-stage sleep (*Medeiros et al., 2021*; *Iglesias et al., 2019*; *Pophale et al., 2023*). Because of these characteristics, cephalopods have been the focus of many fundamental studies by biologists, biophysicists, and physiologists over the past century (*Young, 1974*; *Young, 1962a*; *Young, 1960*; *Young, 1991*; *Young, 1963b*; *Young, 1962b*; *Nixon and Young, 2003*; *Case et al., 1972*; *Dilly et al., 1963*; *Stephens and Young, 1969*). Thanks to advances in sequencing technologies, coleoid cephalopods are now also emerging as animal models in molecular neurobiology, evolution, and genomics (*Baden et al., 2023*). Recent studies examined cephalopod biology from the perspectives of single-cell gene expression (*Styfhals et al., 2022*; *Songco-Casey et al., 2022*; *Duruz et al., 2023*; *Gavriouchkina et al., 2025*), genome topology and gene regulation (*Schmidbaur et al., 2022*; *Rouressol et al., 2023*).

Despite recent technical progress in genomics, the genomes of cephalopods remain challenging to assemble because of their large sizes and high repeat fractions. While the genomes of Octopodiformes (Octopus, Eledone, Argonauta) are either smaller than (1.1 Gigabases or Gb *Yoshida et al., 2022*) or comparable in size to that of humans (around 3 Gb *Albertin et al., 2015*; *Destanović et al., 2023*), the typical genomes of Decapodiformes (squids and cuttlefish) often reach 6 Gb (*Belcaid et al., 2019*; *Albertin et al., 2022*). The biggest contributing factors to this genome expansion are transposable elements (TEs), with different TE classes differentiating squid and cuttlefish genomes from those of octopods (*Marino et al., 2022*). Besides such differences in genome sizes, karyotypes in coleoid cephalopods are also unusual when compared to those of other invertebrates, including other mollusks: for example, haploid chromosome numbers are around 30 in octopuses and around 46 in squids and cuttlefish (*Albertin et al., 2022*; *Simakov et al., 2022*), while they are between 8 and 19 in non-cephalopod mollusks (*Zou et al., 2024*; *Chen et al., 2025*; *Zeng et al., 2021*; *Sun et al., 2024*; *Männer et al., 2024*; *Ma et al., 2023*; *Liu et al., 2023*; *Peñaloza et al., 2021*). Karyotype reconstruction and validation techniques in animals with such large genomes are also not as developed as they are for more familiar invertebrate and vertebrate species.

The common cuttlefish *Sepia officinalis* has been used as a model organism for studies in neural control and development (*Arias-Montecino et al., 2024*; *O'Brien et al., 2017*), behavior (*Hanlon and Messenger, 2018*; *Schnell et al., 2021*), evolution (*Bellingham et al., 1998*), environmental studies (*Chemello et al., 2022*; *Court et al., 2024*), and biomechanics (*Gladman and Askew, 2023*; *Yang et al., 2020*). Found in the Mediterranean, the English Channel, and the European coastal Atlantic, it can produce large egg clutches containing hundreds of embryos, several times throughout its semelparous reproductive cycle (*Boletzky, 1987*; *Laptikhovsky et al., 2003*). More recently, this species has emerged as an important model to study the neural basis of cephalopod camouflage (*Reiter et al., 2018*; *Woo et al., 2023*) building on extensive behavioral studies (*Hanlon and Messenger, 2018*; *Josef et al., 2016*; *Osorio et al., 2022*; *Marshall and Messenger, 1996*; *Zylinski et al., 2009a*; *Zylinski et al., 2012*; *Zylinski et al., 2009b*) and neuronal cell type characterization, evolution, and development (*Imarazene et al., 2017*; *Cocci et al., 2023*; *Andouche et al., 2013*).

Despite a widespread interest in this animal, including the recent sequencing of several cephalopod species as part of the Aquatic Symbiosis Genomics project (*McKenna et al., 2021*), a detailed, annotated genomic resource for this species is still lacking. We describe here the sequencing, assembly, and annotation of the *Sepia officinalis* genome.

Our data provided initial evidence for the existence of 1n=47 chromosomes. We compared them with available genome assemblies (i) from another *S. officinalis* individual by the Darwin Tree of Life

project (DToL) (**Blaxter et al., 2022**), itself suggesting 49 chromosomes, and (ii) from other decapod species, such as *Euprymna scolopes, Doryteuthis pealeii,* and *Acanthosepion esculentum,* each with estimates of 46 chromosomes (**Albertin et al., 2022**). When compared with the assemblies generated for the other species, both *S. officinalis* assemblies contained additional and independent chromosome splits. We thus investigated the repeat content and the alignment of raw data at the discrepant chromosome junctions to assess whether these differences were of a technical or biological nature.

## Results

We sequenced the genome of a 6-month-old *Sepia officinalis* male individual, reared from eggs collected in the Portuguese Atlantic coast. The DNA was extracted from brain tissue and sequenced using long-read (PacBio HiFi) and chromatin conformation (Hi-C) methods (**Figure 1A and B**). The sex of the animal was confirmed by qPCR following a recently described protocol (**Rubino et al., 2025**).

### Genome size and heterozygosity

The haploid genome size of *S. officinalis* was estimated to be ~5.14 Gb based on k-mer estimation from the short-read data, with a high repeat content of 54% (**Ranallo-Benavidez et al., 2020**; **Vurture et al., 2017**). The heterozygosity rate was estimated to be 1.03%, which is higher than in octopus genomes, yet moderate among marine invertebrates (**Albertin et al., 2015**; **Destanović et al., 2023**; **Albertin et al., 2022**).

The size of the scaffolded assembly was 5.68 Gb (**Figure 1C**), about 10% greater than our initial GenomeScope estimate. Indeed, the sizes of most published metazoan genome assemblies deviate by more than 10% from estimates, with proportionally greater deviations as assembly size increases (**Hjelmen, 2024**).

Completeness of the genome was assessed using Benchmarking Universal Single-Copy Orthologs (BUSCO **Simão et al., 2015**) with *metazoa_odb12*. It was 94.3% [single copy: 93.2%, duplicated: 1.2%], with 4.5% fragmented and 1.2% missing BUSCOs. The BUSCO score with *mollusca_odb12* for the final assembly was 90.2% completeness [single copy: 89.1%, duplicated: 1.1%] with 4.9% fragmented and 4.9% missing BUSCOs.

### Assembly-based karyotype

The assembly was generated from PacBio HiFi reads and chromosome conformation capture (Hi-C) reads with ca. 23-fold and 11-fold coverage, respectively. Scaffolding was performed using Hi-C reads, placing 95.8% of bases in 47 scaffolds. The Hi-C heatmap in **Figure 1D** shows the chromatin contacts in clusters corresponding to these 47 scaffolds, suggesting that the 47 clusters each correspond to a single chromosome.

To further test the quality of our assembled *S. officinalis* karyotype, we used several scaffolding programs and manually curated the scaffolds using two independent approaches. The first rested on HapHiC (**Zeng et al., 2024**), a tool based on Hi-C data that uses allele information from primary assembly programs, and scaffolds assemblies with the constraint of an expected number of chromosomes. The resulting Hi-C contact maps with an input of 46, 47, 48, 49, and 50 (**Figure 1—figure supplement 1**) supported 47 scaffolds: an input of 46 scaffolds resulted in one clear chromosome merger (blue arrowhead); higher input numbers resulted in false chromosome splits (black arrowheads, 48, 49, and 50 scaffolds).

Second, we manually curated the scaffolds (obtained with YAHS **Zhou et al., 2023**) using JBAT (**Dudchenko et al., 2018**). These two approaches, conducted independently and both based on Hi-C signal, again converged on 47 chromosomal scaffolds.

### Karyotype comparisons with other Decapods

Several chromosome-scale genome assemblies of decapod cephalopods have been published recently. From those studies, a haploid karyotype of 46 chromosomes seems to be common and conserved among Decapodiformes (**Gavriouchkina et al., 2025**; **Albertin et al., 2022**; **Sanchez et al., 2026**; Figure 3A). Thus, we sought to further investigate and confirm our estimated and different karyotype for *S. officinalis.*

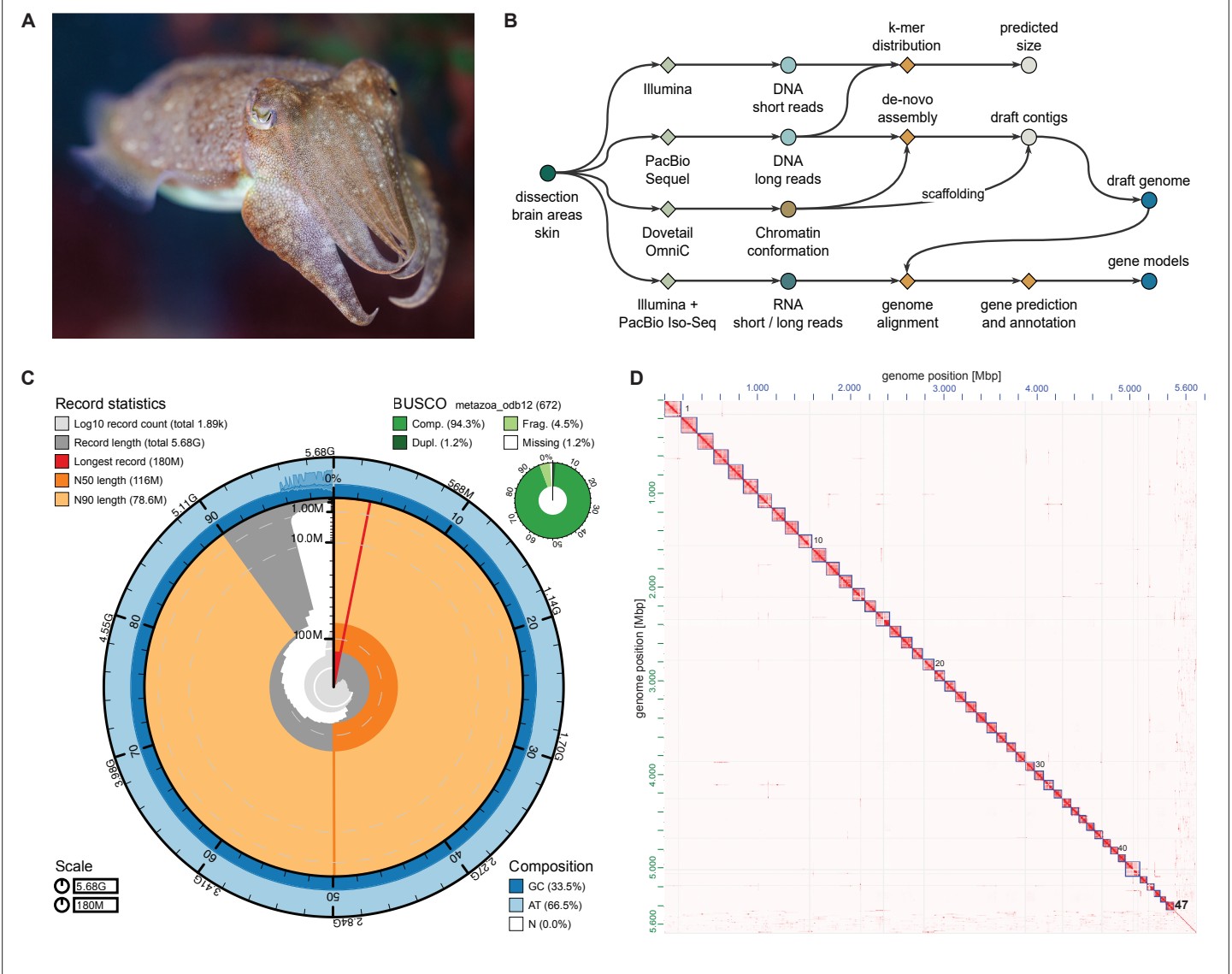

**Figure 1.** *Sepia officinalis* assembly statistics and quality control. (**A**) Specimen of *S. officinalis* (credit: Stephan Junek, MPI for Brain Research). (**B**) Overview of the genome assembly workflow. Genome size was estimated from short DNA reads (Illumina) using GenomeScope (***Ranallo-Benavidez et al., 2020***; ***Vurture et al., 2017***). The primary assembly was generated from long DNA reads (PacBio Sequel II) and chromatin conformation capture (Hi-C) reads (Dovetail OmniC) with hifiasm (***Cheng et al., 2021***). Assembly was scaffolded with YAHS (***Zhou et al., 2023***) and residual small scaffolds were manually placed in chromosomes. (**C**) Snail plot of chromosome-scale *S. officinalis* assembly generated using blobtools2 (***Challis et al., 2020***) showing scaffold statistics (e.g. number of scaffolds, median scaffold length N50), base composition, and completeness measured using Benchmarking Universal Single-Copy Orthologs (BUSCO) (***Simão et al., 2015***) against the *metazoa_odb12* database. (**D**) Hi-C heatmap showing the 47 chromosome-scale scaffolds with few sequences remaining in unplaced scaffolds. X and y-axes show the genome position in Mbp. The heatmap was generated using juicebox (***Dudchenko et al., 2018***), 0–7039 observed counts (balanced) are shown.

The online version of this article includes the following figure supplement(s) for figure 1:

**Figure supplement 1.** HapHiC scaffolding for different numbers of expected chromosome scaffolds show 47 chromosomes as most supported.

Besides assembly-based karyotypes, several cephalopod species have been investigated using cytogenetic techniques. These studies, however, report widely varying estimates of chromosome numbers for decapods, reflecting the difficulty of resolving large (diploid) chromosome numbers in situ. For example, 1n=46 chromosomes has been reported for two species of cuttlefish (*Acanthosepion esculentum* and *Acanthosepion lycidas*) and three loliginid squids (***Gao and Natsukari, 1990***); while 1n=34 chromosomes has been reported for *Aurosepina arabica* (***Jazayeri et al., 2011***) and

**Table 1.** Statistics of *S. officinalis* assemblies from two independent datasets, assembled using a common pipeline.

| version | MPIBR reassembly | | | DToL reassembly | | |
|---|---|---|---|---|---|---|
| | p_ctg | hap1 | hap2 | p_ctg | hap1 | hap2 |
| number of contigs | 8.289 | 10.651 | 10.425 | 8.783 | 11.026 | 11.089 |
| raw length [bp] | 6.049.669.443 | 5.675.386.986 | 5.662.586.038 | 6.053.996.452 | 5.721.157.269 | 5.950.565.264 |
| N50 length [bp] | 1.723.203 | 1.032.632 | 1.010.375 | 1.810.137 | 1.165.578 | 1.182.649 |
| average contig length [bp] | 729.843 | 532.850 | 543.173 | 689.285 | 518.878 | 536.618 |

1n=24 chromosomes in *Acanthosepion pharaonis* (**Ebrahimi Pour, 2009**). In *Sepia officinalis*, a karyotype of 1n=52 has even been described from testis samples (**Vitturi et al., 1982**).

## Comparison with another chromosome-scale *Sepia officinalis* assembly

A chromosome-scale assembly for *Sepia officinalis* was released recently by the Wellcome Sanger Institute's Darwin Tree of Life project (**Blaxter et al., 2022**) (DToL, GCA_964300435.1). That genome was assembled from a male individual using high coverage PacBio Sequel II (~51 x) and Arima2 Hi-C (~80 x) data, with a final assembly size of 5.8 Gb. The haploid chromosome number was estimated to be 49. To compare both *S. officinalis* datasets directly, we downloaded the DToL data and created two new assemblies using the pipeline described above (hifiasm using PacBio HiFi and Hi-C data). The resulting assemblies were overall very similar, with the DToL assembly having a slightly higher contiguity (N50 length, see *Table 1*) and BUSCO completeness (*Figure 2—figure supplement 1A and B*) due to their higher sequencing coverage.

After scaffolding with YAHS, both datasets reached the previously identified chromosome numbers (1n=47 for MPIBR and 1n=49 for DToL, *Figure 2A and B*). To further investigate this surprising discrepancy, we aligned both assemblies using Winnowmap (*Jain et al., 2022*; *Jain et al., 2020*) to locate the differences between them (*Figure 2C*). We observed four 'breakpoints' (BP) of chromosome scaffolds: one in the MPIBR assembly compared to DToL (BP1: DToL_5=MPIBR_40+44) and three in the DToL assembly compared to MPIBR (BP2: DToL_31+40 = MPIBR_2, BP3: DToL_41+46 = MPIBR_6, BP4: DToL_44+45 = MPIBR_7). We also aligned the assemblies to the chromosome-scale genome of another cuttlefish *Acanthosepion esculentum* (1n=46, GCA_964036315.1). In this alignment, all four breakpoints were collinear with single *A. esculentum* chromosomes (*Figure 2D*).

To better understand the potential cause of these divergent chromosome numbers, we analyzed the Hi-C and HiFi coverage in the breakpoint regions (*Figure 2—figure supplement 2A*). First, we aligned the Hi-Fi reads to the scaffolds and extracted all alignments along the 200 kb terminal scaffold windows to find any notable drops in coverage, or reads spanning any of the scaffold junctions. We detected no spanning reads. This is not surprising given that no contigs were assembled at these sites, resulting in the observed scaffold junctions. More interestingly, we noted a~5 fold decrease in HiFi coverage along the DToL scaffold_40 (part of BP2) relative to its flanking regions, indicating a highly repetitive, low-mappability region at this boundary.

Next, we realigned the Hi-C data to the scaffolded assemblies using bwa-mem2 (*Vasimuddin et al., 2019*) and extracted all trans HiC pairs (between-scaffold contacts) using pairtools (*Abdennur et al., 2023*). We normalized trans HiC contacts to the scaffold length and compared contact rates between breakpoint scaffolds to the baseline contact rate (computed from pairs of scaffolds with a clear 1-to-1 match between assemblies), and the contact rate within scaffolds (intra-scaffold pairs) (*Figure 2—figure supplement 2B and C*). The contact rates within breakpoints were consistently lower than within scaffolds, likely falling below the threshold to be merged during assembly. However, the contact rates at three of four breakpoints (BP1, BP3, BP4) were significantly elevated above the genome-wide background distribution (empirical $p=0.010$, 0.005, 0.005, respectively), suggesting that they may represent intra-chromosomal contacts disrupted by a misassembly. Notably, BP2 was not significant (empirical $p=0.170$), likely due to the low coverage and mappability around the DToL scaffold_40 boundary. Considered jointly, the three DToL breakpoint scaffold pairs showed significantly higher trans contact rates than the background (Wilcoxon rank-sum, one-tailed, $U=1771$, $p=0.004$).

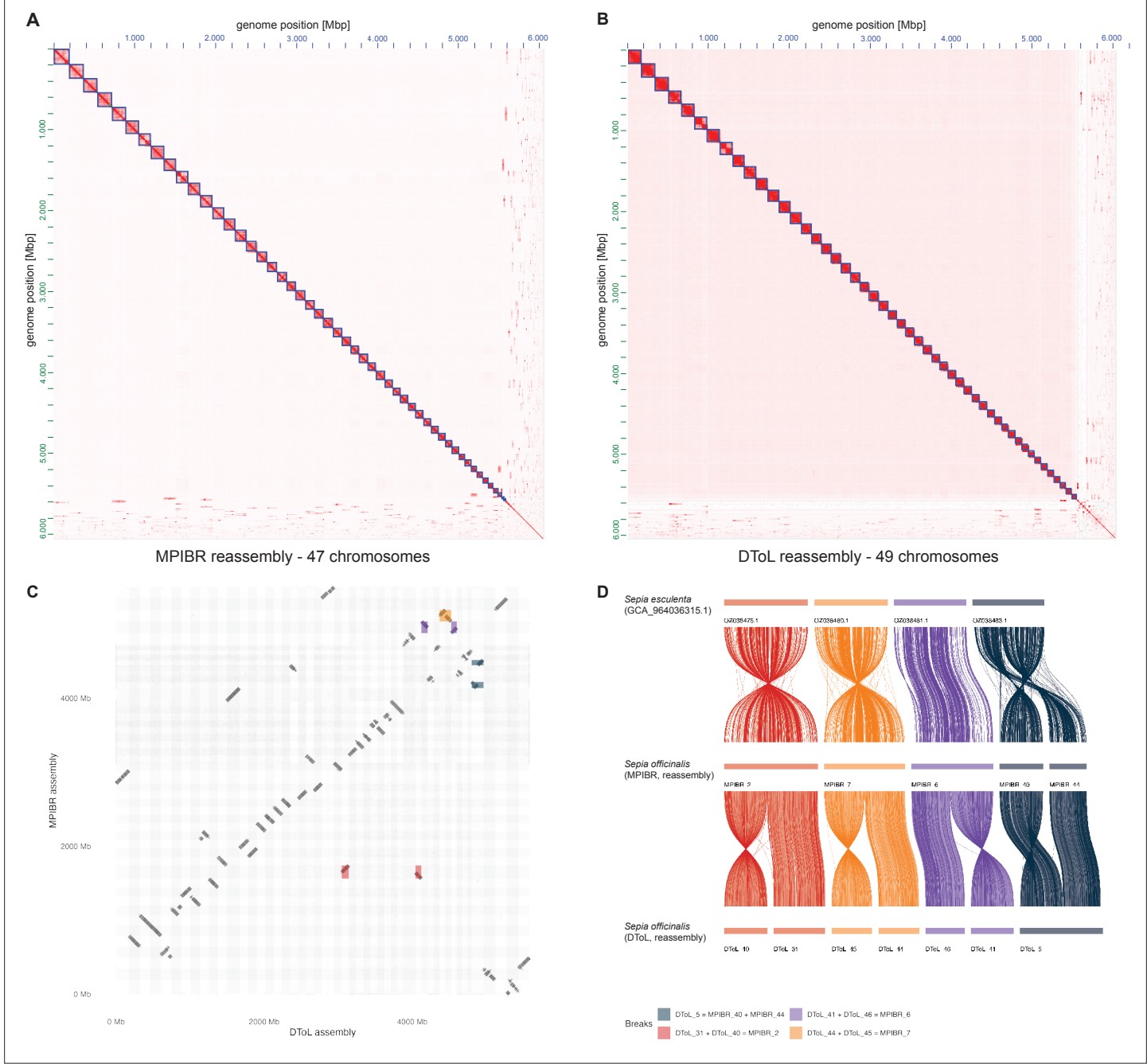

**Figure 2.** Comparison of two *Sepia officinalis* chromosome-scale assemblies indicates chromosome number of 1n=46. Datasets were collected from two *S. officinalis* animals, one as described in this study (MPIBR), the second by the Darwin Tree of Life consortium (DToL) (**Blaxter et al., 2022**). Both datasets were assembled using a common pipeline (hifiasm and YAHS). (**A**) Hi-C contact map of the MPIBR primary assembly, scaffolded using YAHS without manual curation. Assembled 47 chromosome scaffolds are shown as blue boxes. (**B**) Hi-C contact map of DToL primary assembly, scaffolded using YAHS without manual curation, showing 49 assembled chromosome scaffolds as blue boxes. (**C**) Whole-genome alignment of both scaffolded assemblies using Winnowmap2 (**Jain et al., 2022**), showing DToL on x-axis and MPIBR on the y-axis. The 4 'breakpoints' of chromosomes in either of the assemblies (three breaks in DToL chromosomes compared to MPIBR, one break in MPIBR compared to DToL) are highlighted in different colors. (**D**) Ribbon diagram showing the four breakpoints from (**C**) compared to the chromosome-scale assembly from another cuttlefish, *Acanthosepion esculentum* (1n=46). The color of breakpoints are the same in panels **C**+**D**.

The online version of this article includes the following figure supplement(s) for figure 2:

**Figure supplement 1.** BUSCO completeness results.

**Figure supplement 2.** Analysis of raw data at breakpoints between *S. officinalis* assemblies hints at a technical cause of breakpoints.

Lastly, we analyzed the repeat landscape around the 200 kb scaffold ends using RepeatMasker (*Smit et al., 2025*) and the custom repeat library that we had generated for *Sepia officinalis* (described further below). Compared to control scaffolds of the same assembly, we observed consistently elevated repeat content at the breakpoint junctions (mean 71.5% vs 67.6% masked bases), with an enrichment of unclassified repeats (32.1% vs 30.0%), which could explain a repeat-driven assembly fragmentation or scaffolding failure (*Figure 2—figure supplement 2D*). The BP2 DToL scaffold_40 junction window was 99.99% masked (99.2% unclassified repeats), providing a likely mechanistic explanation for both the HiFi coverage drop and the absence of a significant trans Hi-C signal at this breakpoint. Taken together, these analyses suggest that the different chromosome numbers across the two *S. officinalis* assemblies are due to technical reasons, caused

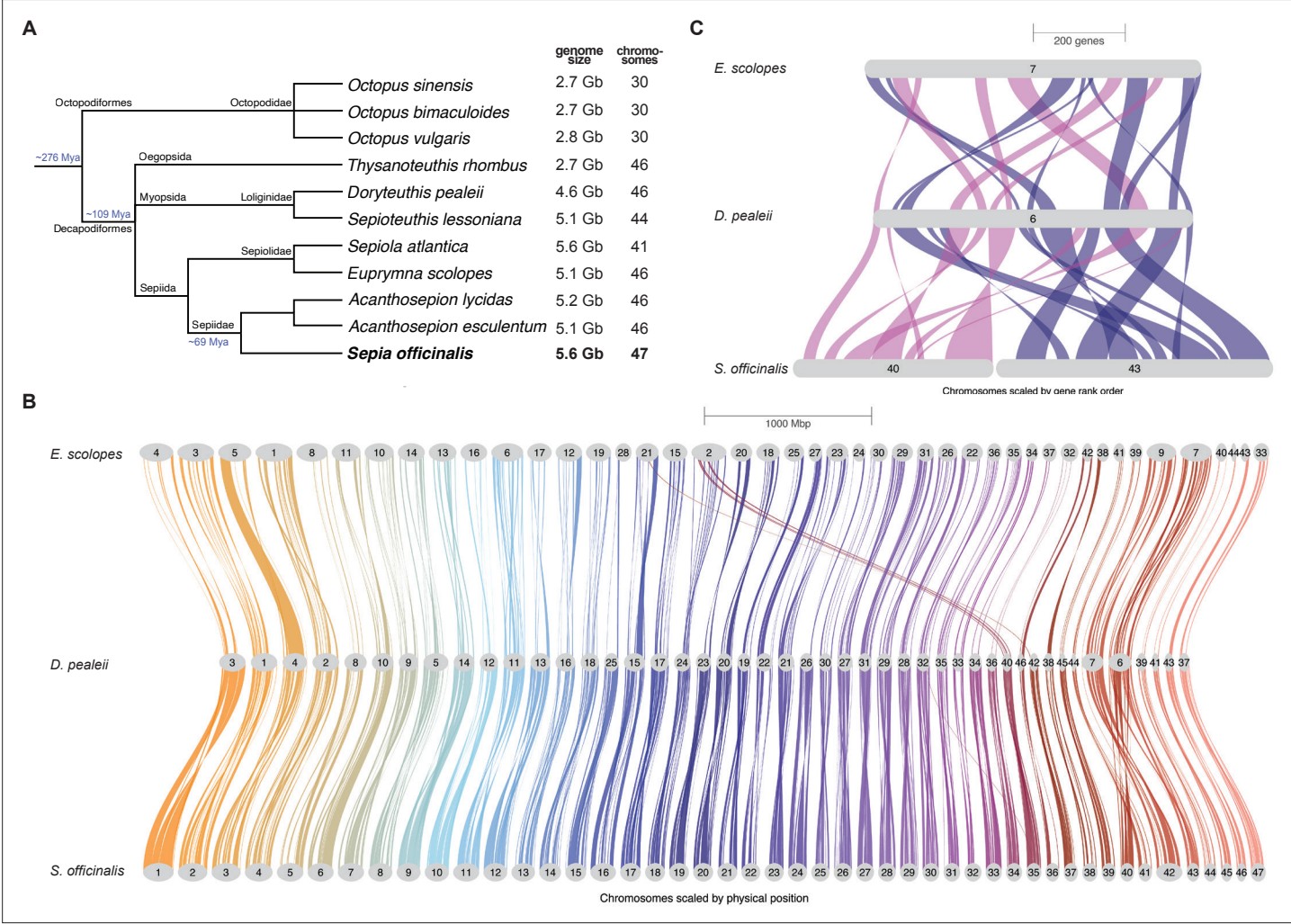

**Figure 3.** Syntenic comparison of three decapod species. (**A**) Taxonomy of selected cephalopod species showing their genome size (in gigabases, Gb) and haploid chromosome numbers. Taxonomy information was downloaded from NCBI taxonomy browser, divergence times for Coleoidea and Decapodiformes from *Kröger et al., 2011* and for Sepiidae from *López-Córdova et al., 2022*. (**B**) Genome-wide syntenic relationship between chromosomes of *E. scolopes* (*Albertin et al., 2022*) (top), *D. pealeii* (*Albertin et al., 2022*) (middle), and *S. officinalis* (bottom). Colored braids connect syntenic regions across genomes, with chromosomes drawn to physical scale. *Euprymna* chromosomes 45 and 46 are not shown because they contain too few orthogroups. (**C**) Detailed synteny of *Sepia* chromosomes 40 (magenta) and 43 (dark blue) shown, that are joined in the other species and cause the different haploid chromosome number in *Sepia*. Riparian plots were generated using GENESPACE v1.2.3 (*Lovell et al., 2022*).

The online version of this article includes the following figure supplement(s) for figure 3:

**Figure supplement 1.** Syntenic relationship between *S. officinalis* and *D. pealeii* chromosomes.

**Figure supplement 2.** Syntenic relationship between *S. officinalis* and *E. scolopes* chromosomes.

**Figure supplement 3.** Syntenic comparison of four decapod species hints at a cephalopod sex chromosome.

by repeat-rich scaffold boundaries that impair HiFi and Hi-C read alignment and in turn, correct assembly in these regions.

## Comparisons with other chromosome-scale Decapodiformes genomes

To further investigate why our initial estimate of *Sepia officinalis's* chromosome numbers differed from those in other studies, we compared our assembly with the chromosome-scale genomes of two other decapod cephalopods, *Euprymna scolopes* (*Schmidbaur et al., 2022*) (a sepiolid) and *Doryteuthis pealeii* (*Albertin et al., 2022*) (a loliginid), both described as having 46 chromosomes (*Figure 3A*). Using orthogroups to perform linkage analysis, we could detect a clear chromosome homology between *Doryteuthis* and *Sepia*, and to a lesser extent between *Euprymna* and *Sepia* (*Figure 3B*). We observed some small-scale rearrangements between *Euprymna* and *Doryteuthis*, such as a fusion of chromosomes 24 and 40 from *Doryteuthis* in chromosome 2 of *Euprymna*, which has also been observed in other Sepiolids (*Gavriouchkina et al., 2025*; *Sanchez et al., 2026*). Dotplots showing detailed pairwise synteny comparisons are shown in *Figure 3—figure supplement 1* (*Sepia* to *Doryteuthis*) and *Figure 3—figure supplement 2* (*Sepia* to *Euprymna*). The inferred species tree as part of this analysis places *Euprymna* as sister to *Sepia* and *Doryteuthis,* matching recent phylogenetic analyses (*Chen et al., 2025*; *López-Córdova et al., 2022*) (note, however, that this tree was constructed without the inclusion of outgroup taxa, and therefore lacks a reliable root).

This comparison revealed that chromosomes 40 and 43 in *Sepia* are merged into one in *Euprymna* and in *Dorytheutis* (*Figure 3C*), as they were in the DToL *Sepia officinalis* assembly (*Figure 2C*). Together with the whole-genome alignments between the cuttlefish assemblies described earlier (*Figure 2D*), these results lead to the following parsimonious conclusion: that the karyotype for *S. officinalis* is 1n=46 chromosomes, as it is for other Decapodiformes.

We also performed a synteny comparison, including the cuttlefish *A. esculentum* that was recently annotated by genome liftover from *Acanthosepion pharaonis* (*Coffing et al., 2024*). That assembly, constructed from a female individual, showed a low read coverage for a sex chromosome in a ZZ/Z0 sex determination system (*Coffing et al., 2025*). Our syntenic comparison indicated a strong homology between the inferred Z chromosome of *A. esculentum* and chromosome 46 of *S. officinalis* (*Figure 3—figure supplement 3A*). As indicated above, we determined the sex of our *S. officinalis* specimen by replicating the analysis used to identify the Z chromosome in *A. esculentum* (*Coffing et al., 2025*). For this, we aligned short-reads (Illumina) from the same *S. officinalis* individual to the assembly and examined the normalized read coverage for each chromosome (*Figure 3—figure supplement 3B*). In contrast to the low coverage observed in the female *A. esculentum* assembly (*Figure 3—figure supplement 3C*), we observed no significant decrease in read coverage for chromosome 46, suggesting that our material came from a male animal. Additionally, we used a recently published genotyping protocol for cephalopods (*Rubino et al., 2025*) and performed qPCR on extracted genomic DNA from tissue samples, confirming the male sex of our sequenced individual.

## Genome repeat landscape

After creating a custom repeat library for *Sepia officinalis* using RepeatModeler (*Flynn et al., 2020*), we masked the genome using RepeatMasker (*Smit et al., 2025*), resulting in 71.17% masked bases. The categories of repeats are shown in *Figure 4A*. Most repeats were not characterized (39.65% of total bp) and presumably represent ancient repeats that diverged beyond recognition (*Feschotte, 2008*). Retroelements constituted the largest characterized repeat category (17.32%) followed by DNA transposons (5.92%); 5.83% were annotated as simple repeats. As observed in other Decapodiformes, the *Sepia* genome contained almost no short interspersed nuclear elements (SINEs), supporting the hypothesis that the SINE expansion observed in octopuses occurred independently in their lineage (*Marino et al., 2022*).

## Gene modeling and annotation

The genome was further annotated using BRAKER (*Simão et al., 2015*; *Hoff et al., 2019*; *Brůna et al., 2021*; *Brůna et al., 2024a*; *Gabriel et al., 2021*; *Hoff et al., 2016*; *Stanke et al., 2006*; *Stanke et al., 2008*; *Li, 2023*; *Iwata and Gotoh, 2012*; *Gotoh, 2008*; *Buchfink et al., 2015*; *Kovaka et al., 2019*; *Brůna et al., 2024b*; *Huang and Li, 2023*; *Pertea and Pertea, 2020*; *Gabriel et al., 2024*) combining short- and long-read RNA-seq data and publicly available protein data from multiple

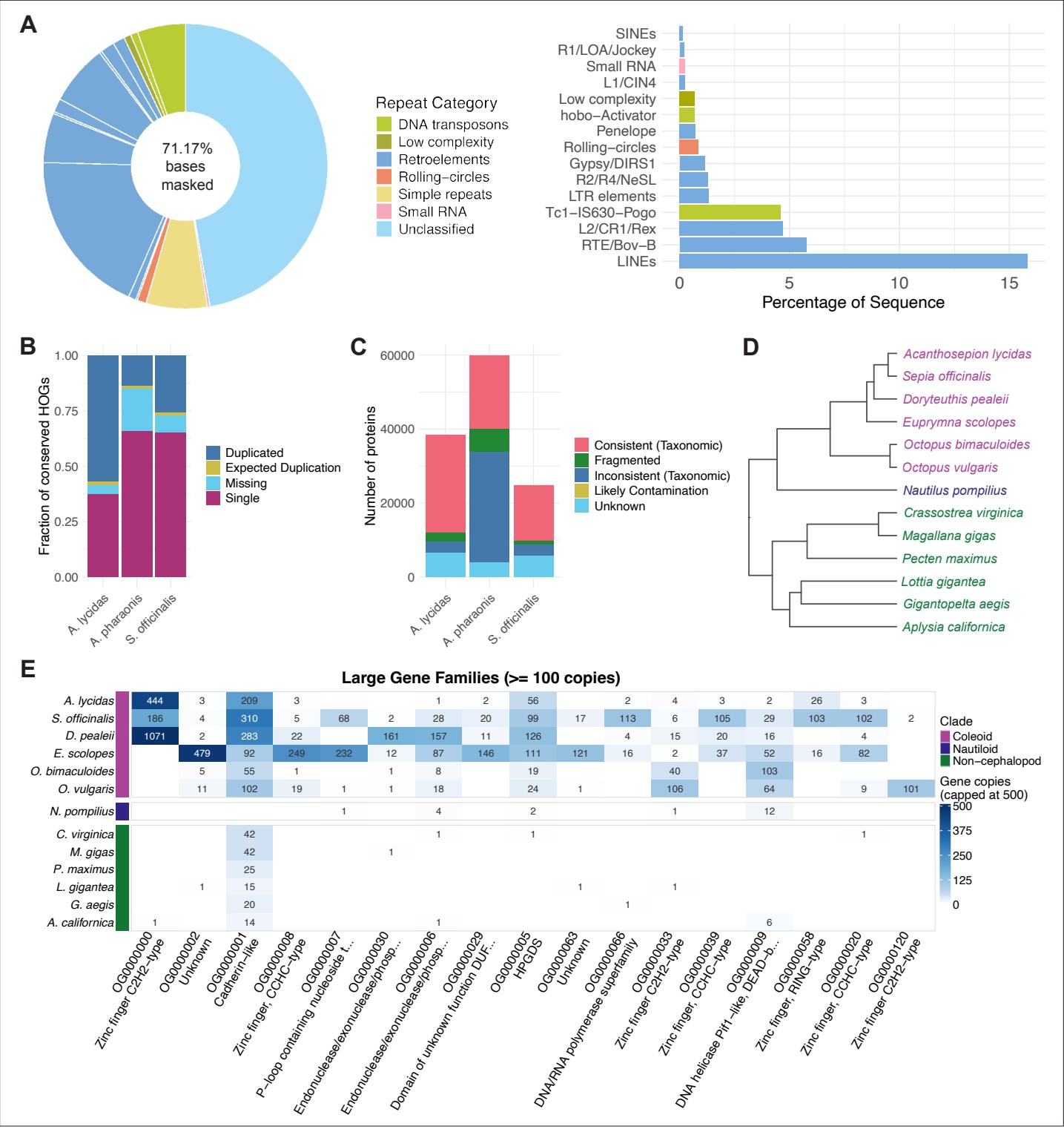

**Figure 4.** Genome annotation for *Sepia officinalis*. (**A**) Annotation of repeat landscape of the *S. officinalis* genome, annotated using RepeatModeler (*Flynn et al., 2020*). Full repeat landscape is shown on the left, annotated repeats (excluding unclassified or simple repeats) are shown on the right. (**B–C**) Quality control of gene annotation and comparison to two other cuttlefish species using OMArk (*Nevers et al., 2022*). Results shown for *Acanthosepion lycidas* (GCA_963932145.1, Ensembl Genebuild), *Sepia officinalis* (BRAKER, this study), and *Acanthosepion pharaonis* (*Song et al., 2021*) (BRAKER). Lophotrochozoa was used as the ancestral clade. (**B**) Completeness assessed by the presence of genes conserved in the clade, classified as *single* or multiple copies (*duplicated*), or *missing*. (**C**) Consistency assessed by the proportion of proteins placed in the correct lineage (*consistent*); placement in incorrect lineages randomly (*inconsistent*) or to specific species (*contamination*), or no placement in known gene families (*unknown*).

*Figure 4 continued on next page*

*Figure 4 continued*

(**D**) Phylogenetic tree of 13 molluscan species used for analysis of gene families with Orthofinder (*Emms et al., 2025*). Species are colored by clade: purple = coleoid cephalopods, blue = nautiloid (non-coleoid cephalopod), green = non-cephalopod mollusk. (**E**) Heatmap of largest gene families (orthogroups from Orthofinder, with more than 100 genes in any species), ordered from largest gene count across all species on the left. Families with at least one gene in *S. officinalis* are depicted. Rows show gene counts for each species (color capped at 500 genes), columns show orthogroups and their annotation by eggNOG mapper (*Cantalapiedra et al., 2021*; *Huerta-Cepas et al., 2017*) or InterProScan (*Blum et al., 2025*), if available. Clade colors match (**D**).

The online version of this article includes the following figure supplement(s) for figure 4:

**Figure supplement 1.** Gene family expansion analysis.

molluscan species (*Doryteuthis pealeii* **Albertin et al., 2022**, *Euprymna scolopes* **Rogers et al., 2024**, *Octopus bimaculoides* **Albertin et al., 2022**, *Octopus vulgaris* **Destanović et al., 2023**, *Nautilus pompilius* **Zhang et al., 2021**, and *Pecten maximus* **Zeng et al., 2021**). A total of 18,663 gene models and 23,768 proteins were annotated.

The gene annotation was evaluated using BUSCO (*Simão et al., 2015*) v5.5.0 in protein mode, showing high completeness of the annotation (*metazoa_odb10*: C:98.2%[S:77.3%,D:20.9%], F:1.0%, M:0.8%, n:954 and *mollusca_odb10*: C:81.5%[S:59.6%,D:21.9%], F:0.9%, M:17.6%, n:5295). We further checked the completeness and consistency of our gene models using OMArk (*Nevers et al., 2022*), and compared them to genome annotations for two other cuttlefish species, *Acanthosepion pharaonis* and *Acanthosepion lycidas* (*Figure 4B*). Completeness was assessed by comparing the annotated genes to conserved orthologs (present in 80% of extant species) in a given taxonomic clade (here, the superorder Lophotrochozoa). The *S. officinalis* annotation missed 181 out of 2373 genes, which is higher than with *A. lycidas* (96 genes) but lower than with *A. pharaonis* (458 genes).

In a consistency assessment, where the presence of known gene families from the lineage is evaluated (*Figure 4C*), *S. officinalis* contains low proportions of taxonomically inconsistent or fragmented proteins (ca. 13%) similar to *A. lycidas* (9%), giving confidence in the annotation. In contrast, more than 50% of *A. pharaonis* proteins are labeled as inconsistent or fragmented, which could indicate annotation errors (*Nevers et al., 2022*).

Overall, the annotations contain different numbers of predicted proteins which may reflect differences in the annotation method and the reference data used. The genome of *S. officinalis* contains fewer proteins (23,768) than those of *A. lycidas* (35,949) or *A. pharaonis* (53,515). In comparison, two octopus genomes contain similar numbers of proteins (*O. vulgaris*: 30,134; *O. bimaculoides*: 29,037) and were produced using NCBI's RefSeq (*Goldfarb et al., 2025*) pipeline, suggesting that the differences observed across cuttlefish proteomes are probably of technical instead of biological origin.

Lastly, we assigned orthology information to the *S. officinalis* proteome using InterProScan (*Blum et al., 2025*) and eggNOG-mapper (*Cantalapiedra et al., 2021*) to aid the interpretability of the resource for future transcriptomic or proteomic studies. Overall, 89% of proteins (21,204 out of 23,768) received an annotation from InterProScan from at least one of their databases. 59% of proteins (14,126 out of 23,768) were annotated by eggNOG-mapper, reflecting the more stringent orthology filters and prioritization of full-length matches implemented in the program (*Huerta-Cepas et al., 2017*).

## Analysis of expanded gene families

We sought to investigate the *S. officinalis* gene annotation and place it in the context of gene repertoires from other cephalopod or molluscan species. First, we collected available genome annotations from 12 other molluscan species (*Table 2*) and clustered them using OrthoFinder v.3.1.0 (*Emms et al., 2025*), resulting in 23,658 orthogroups, hereafter named gene families.

First, we investigated 36 of the gene families that contain more than 100 genes in any of the species, with 17 of these families containing at least one gene of *S. officinalis*, that reflect large-scale gene family expansions (*Figure 4E*). We used the InterProScan and eggNOG-mapper annotations to infer functional roles of these genes, selecting the most common gene annotation as the name of the gene family.

The zinc finger C2H2-type transcription factors (TFs) were grouped into three of the large gene families, with the largest family (OG0000000) only present in decapod cephalopods. This likely reflects the largely independent expansions in the octopod and decapod lineages that date back to a burst

**Table 2.** Overview of gene annotation of 13 molluscan species used for gene family analysis.

| Organism Scientific Name | Accession | Source | URL | # of Proteins |
|---|---|---|---|---|
| *Aplysia californica* | GCF_000002075.1 | RefSeq | https://www.ncbi.nlm.nih.gov/datasets/genome/GCF_000002075.1/ | 21897 |
| *Crassostrea virginica* | GCF_053477285.1 | RefSeq | https://www.ncbi.nlm.nih.gov/datasets/gene/GCF_053477285.1/ | 53819 |
| *Doryteuthis pealeii* | GCA_023376005.1 | custom (*Albertin et al., 2022*) | https://metazoa.csb.univie.ac.at/CephData/dorPea.prot.gz | 24931 |
| *Euprymna scolopes* | GCA_024364805.1 | Github (*Rogers, 2025*) | https://github.com/TheaFrances/E.scolopes-V2.2-BRAKER2-gene-annotation | 31908 |
| *Gigantopelta aegis* | GCF_016097555.1 | RefSeq | https://www.ncbi.nlm.nih.gov/datasets/genome/GCF_016097555.1/ | 24904 |
| *Lottia gigantea* | GCF_000327385.1 | RefSeq | https://www.ncbi.nlm.nih.gov/datasets/genome/GCF_000327385.1/ | 23822 |
| *Magallana gigas* | GCF_963853765.1 | RefSeq | https://www.ncbi.nlm.nih.gov/datasets/genome/GCF_963853765.1/ | 35231 |
| *Nautilus pompilius* | GWHBECW00000000 | GWH | https://ngdc.cncb.ac.cn/gwh/Assembly/21849/show | 16536 |
| *Octopus bimaculoides* | GCF_001194135.2 | RefSeq | https://www.ncbi.nlm.nih.gov/datasets/genome/GCF_001194135.2/ | 29037 |
| *Octopus vulgaris* | GCA_951406725.2 | RefSeq | https://www.ncbi.nlm.nih.gov/datasets/gene/GCA_951406725.2/ | 30134 |
| *Pecten maximus* | GCF_902652985.1 | RefSeq | https://www.ncbi.nlm.nih.gov/datasets/genome/GCF_902652985.1/ | 28975 |
| *Acanthosepion lycidas* | GCA_963932145.1 | Ensembl genebuild | https://ftp.ebi.ac.uk/pub/ensemblorganisms/Sepia_lycidas/GCA_963932145.1/ensembl/geneset/2024_05/ | 35949 |
| *Sepia officinalis* | GCA_050097725.1 | this study | https://doi.org/10.17617/1.5n7h-4385 | 23768 |

of transposon activity ca. 25 million years ago (*Albertin et al., 2015*; *Belcaid et al., 2019*; *Albertin et al., 2022*). The largest expansion across mollusks occurs in the cadherin-like family (OG0000001): 310 in *S. officinalis*, 283 in *D. pealeii*, 209 in *A. lycidas*, 102 in *O. vulgaris*, 55 in *O. bimaculoides*, with low but non-zero counts in bivalves (*C. virginica*, *M. gigas*). This profile is consistent with the proto-cadherin expansion first described in *O. bimaculoides* (*Albertin et al., 2015*) and subsequently shown to be present across cephalopods (*Belcaid et al., 2019*; *Albertin et al., 2022*; *Styfhals et al., 2019*).

HPGDS (OG0000005, hematopoietic prostaglandin D synthase) is a glutathione-S-transferase family member that catalyzes the conversion of prostaglandins, which have well-described roles in immune responses in vertebrates and insects (*Ahmed et al., 2018*; *Kanaoka and Urade, 2003*). This family shows a broad expansion in decapods, with a lesser expansion in octopods. Additionally, members of the glutathione-S-transferase families have been co-opted as S-crystallins, structural proteins found in the lens of cephalopods that may, or may not, retain enzymatic functions (*Tomarev et al., 1995*; *Tan et al., 2016*).

Two large families are mostly lineage-restricted. The RING-type zinc finger family (OG0000058) has 103 copies in *S. officinalis* and 26 in *A. lycidas* but is absent in all other species except for *E. scolopes*. Conversely, OG0000002 (unknown function) has 479 copies in *E. scolopes* and only a few copies in the other species. This interesting Sepiolid-specific expansion warrants further characterization.

We estimated gene family evolution rates using CAFE5 (*Mendes et al., 2021*) for all families with less than 100 copies in any species (this excludes the families described above, as very large copy-number differences between species preclude likelihood calculations under the applied birth-death model). After comparing different model parameters, we chose a gamma model with three rate categories, allowing for evolutionary rate variation among gene families. Out of the 12,895 gene families analyzed, 1813 showed a significant ($p<0.05$) expansion or contraction in at least one of the species. We focused our analysis on the 30 most significantly expanded families; among them were several retrotransposon-associated domains that have expanded specifically in *S. officinalis*: five families

carrying Retrovirus-related Pol polyprotein domains, two Reverse transcriptase domain families, and four Ribonuclease H-like families (*Figure 4—figure supplement 1A*). There was no coordinate-based overlap of the coding sequences with annotated TEs from the RepeatMasker output (Methods).

In addition to the three large gene families of C2H2 zinc finger expansions, 45 gene families containing this TF type showed a significant change in the CAFE5 analysis. Notably, eight of the significant gene families, as well as four of the largest gene families, were annotated as CCHC-type zinc fingers, which contain a 'zinc knuckle' motif that is characteristic of retroviral nucleocapsid proteins (*Summers et al., 1992*) and is functionally integrated in the genomes of several species, including humans (*Aceituno-Valenzuela et al., 2020*).

Some gene families without any relationship to retrotransposons were also expanded. For example, the UGT2A1-related family is a UDP-glucuronosyltransferase, a class of enzymes central to phase II

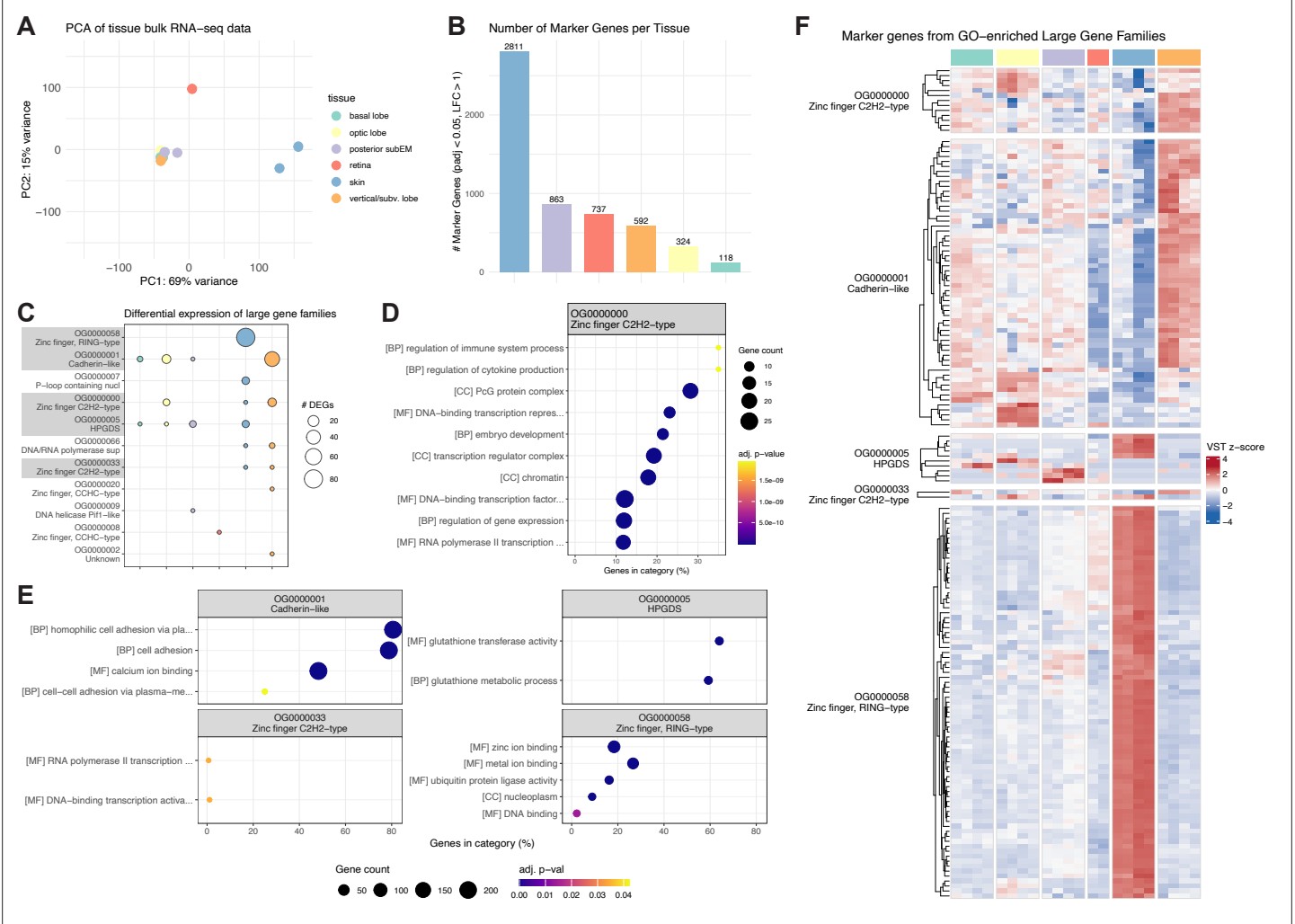

**Figure 5.** Expression of expanded gene families in tissue bulk RNA-seq data. Bulk RNA-seq data collected from one adult *S. officinalis* from different brain tissues (optic lobes - yellow, basal lobes - turquoise, vertical and subvertical lobes - orange, posterior subesophageal mass - purple), retina (red), and skin (blue, from the dorsal mantle). Tissue color code is identical throughout the figure. (**A**) Principal component analysis (PCA) of the data, showing the first 2 PCs, colored by tissue. (**B**) Barplot showing number of differentially expressed (DE) genes (i.e. marker genes) for each tissue, calculated against all other tissues using DESeq2 (*Love et al., 2014*). (**C**) Largest gene families (orthogroups) with differential expression in bulk RNA-seq data. Dot size shows the number of DE genes for each tissue. Families with enriched gene ontology (GO) terms are highlighted in gray. (**D+E**) Dotplots of enriched gene ontology (*Aleksander et al., 2026*; *Ashburner et al., 2000*) (GO) terms for large gene families, enriched using clusterProfiler (*Xu et al., 2024*) using a hypergeometric test. Dot size shows the number of expressed genes per family with this GO term, x-axis shows the percentage of expressed genes from all genes with this GO term. Dot color shows the adjusted p-value after Benjamini-Hochberg false discovery rate (FDR) correction. CC: cellular component, MF: molecular function, BP: biological process. (**F**) Heatmap of z-scored expression of all DE genes from the largest gene families with enriched GO terms.

detoxification and conjugation of metabolites, reported in other mollusks in the context of environmental chemical tolerance (*Qi et al., 2026*), and in insects in the context of pigmentation (*Ahn and Marygold, 2021*). We also detected a family of homeodomain-like proteins, representing an expansion of this important TF family.

## Tissue-specific expression of expanded gene families

To place the identified gene families in a functional context, we profiled their expression in the bulk RNA-seq data (taken from multiple tissues of *S. officinalis*) used originally for gene modeling (*Figure 5A*). Principal component analysis (PCA) revealed the largest axis of variation in gene expression to separate brain tissues from peripheral tissues, with skin being the most transcriptomically distinct (*Figure 5A*), consistent with the high number of tissue-specific differentially expressed (DE) genes identified in non-neural tissues (*Figure 5B*). We identified the genes belonging to expanded families that were differentially expressed across tissues and enriched gene ontology (*Aleksander et al., 2026*; *Ashburner et al., 2000*) (GO) terms for them to gain additional insight. The large families excluded from CAFE5 modelling and the significantly expanded families identified by CAFE5 were analyzed separately.

Eleven of the largest gene families were expressed in our data (*Figure 5C*) and five had enriched GO terms (*Figure 5D and E*). Among them, the cadherin family showed brain-restricted expression and GO terms related to cell–cell adhesion and calcium binding, consistent with their role in neuronal connectivity and circuit formation (*Albertin et al., 2015*; *Redies, 1997*). Two C2H2 zinc finger gene families were expressed in the optic and vertical/subvertical lobes of the brain and in the skin, with GO terms related to DNA-binding, transcriptional regulation, or development. The RING-type zinc finger family was expressed specifically in the skin, with GO terms, including zinc binding and ubiquitin protein ligase activity, the canonical function of RING-domain E3 ligases (*Joazeiro et al., 1999*). Genes of the HPGDS/S-crystallin family were expressed in the brain (basal and optic lobes and posterior subesophageal mass) and skin, with GO terms related to glutathione metabolism, matching their described enzymatic function. We did not find expression in the retina, which is expected given that S-crystallins are expressed in lentigenic cells of the eye (*Gavriouchkina et al., 2025*; *Ryu et al., 2023*) and these cells were not included during sampling.

Among the 30 most significantly expanded families examined (out of 1813 total), expression was widespread (20/30) and tissue-specific differential expression was common (17/30), suggesting that a substantial proportion of expanded paralogs represent functional coding sequences with specialized spatial deployment (*Figure 4—figure supplement 1B*). Ten of the retrotransposon-associated families were differentially expressed in the brain (optic and vertical/subvertical lobes) and skin, arguing against these loci being inactive repeat fragments and supporting their inclusion as transcribed gene models. Two significantly expanded families showed both differential expression and enriched GO terms (*Figure 4—figure supplement 1C*). The first was the UGT2A1-related family, which had the largest number of differentially expressed genes overall, with expression concentrated in the skin, retina, and posterior subesophageal mass of the brain. Enriched GO terms matched the described enzymatic function for this family, namely UDP-glycosyltransferase activity. The second gene family was the homeodomain-like family with enrichment for DNA binding terms consistent with their role as transcription factors, and was preferentially expressed in the vertical and subvertical brain lobes with weaker expression in other areas.

Collectively, many differentially expressed genes from expanded families were restricted to specific tissues or brain subregions (*Figure 5F*, *Figure 4—figure supplement 1D*), indicating that paralogs within an expanded family have adopted distinct spatial expression domains and possibly, specialized functions.

## Discussion

This study presents a detailed, chromosome-scale genome assembly and annotation for the common cuttlefish *Sepia officinalis*, providing an additional important resource for comparative genomics, neurobiology, and evolutionary studies within cephalopods and mollusks. The assembly was derived from a combination of PacBio HiFi long reads, Dovetail Omni-C chromatin conformation capture, and multiple rounds of scaffolding and manual curation. The resulting assembly has a haploid genome size

of 5.68 Gb, and contains 47 chromosome-like clusters. This number prompted us to perform detailed analyses of chromosome structure and homology across various decapod genomes, leading to a likely consensus of 46 chromosomes. This resource should be valuable for future efforts in cephalopod genomics, especially within the Decapodiformes, where large and repetitive genomes have hindered previous efforts.

## Chromosome number and karyotypic variation

Our assembly contains an estimated haploid chromosome number of 47, a result produced by two independent scaffolding approaches (YAHS *Zhou et al., 2023* and HapHiC *Zeng et al., 2024*) and supported by Hi-C contact maps. This karyotype, however, differed from those reported for other decapod cephalopods, such as *Euprymna scolopes* and *Doryteuthis pealeii* (*Albertin et al., 2022*), each with 46 chromosomes. Syntenic comparisons across datasets revealed that chromosomes 40 and 43 in *S. officinalis* correspond to a single chromosome in these other species, suggesting that the apparent split in *Sepia* may result from a technical artefact.

We also compared our results to the recently published *S. officinalis* genome from the Darwin Tree of Life project (DToL) (*Blaxter et al., 2022*), which proposed 49 chromosomes for that species. This difference was surprising: two genome assemblies from the same species are usually expected to have the same karyotype. The discrepancy is attributable to one chromosome split in our assembly and three splits in the DToL assembly, and could stem from technical differences of the Hi-C methods (Dovetail Omni-C vs. Arima v2), sequencing depth (11-fold vs. 83-fold coverage) or the tissue used (optic lobe vs. eye). We investigated these splits, or breakpoints in two ways: first, we reassembled both datasets with the same parameters and aligned both to the genome of another cuttlefish *A. esculentum* (1n=46). In this alignment, all four breakpoints were collinear with single *A. esculentum* chromosomes, providing a phylogenetically grounded prediction of 1n=46 for *Sepia officinalis*. Second, we investigated read depth at the breakpoint junctions, and saw a significant increase of Hi-C read pairs spanning the junctions compared to random chromosome pairings. Together with a higher repeat content at the breakpoints, these analyses suggest a technical cause for the differences between assemblies, caused by repeat-rich scaffold boundaries that impaired HiFi and Hi-C read alignment and, in turn, made correct assembly challenging in these regions.

In conclusion, the two independent genome assemblies and our analysis suggest three possible karyotypes for *Sepia officinalis*: 46, 47, or 49. The most likely explanation based on chromosome synteny is that the true number is 46, matching the numbers established for other cuttlefish and decapod species (*Gavriouchkina et al., 2025*; *Albertin et al., 2022*; *Sanchez et al., 2026*). 47 chromosomes (our initial estimate) and 49 chromosomes (DToL estimate) would, therefore, be erroneous values explained by technical factors related to repeat content and low alignment rates preventing complete assembly. Incorporating ultra-long read data may help to correctly assemble these problematic regions, as is now common for telomere-to-telomere assemblies (*Li and Durbin, 2024*).

Another, but less likely, explanation for the observed differences could be variations in chromosomal architecture of different *Sepia officinalis* populations, due to idiosyncratic individual chromosome fusion. Intraspecific chromosome number variation is rare but not unprecedented; for instance, chromosomal polymorphism has been described in the butterfly *Leptidea sinapis* across different populations (*Lukhtanov et al., 2011*; *Lukhtanov et al., 2018*). Notably, the two *S. officinalis* individuals used for sequencing originated from different regions - the Portuguese Atlantic (this study) and the French Mediterranean (DToL project), raising the possibility of geographic variation in chromosome structure. Future population-level analyses will hopefully determine whether *S. officinalis* exhibits karyotypic polymorphism.

Additional methods such as cytogenetic karyotyping or optical mapping such as BioNano (*Lam et al., 2012*) (imaging of fluorescently tagged, linearized DNA) could be used to validate chromosome numbers. However, whereas karyotypes of octopuses have been consistent throughout the literature (1n=30) (*Wang and Zheng, 2017*; *Adachi et al., 2014*), those measured in decapods vary greatly. For example, 1n=46 chromosomes have been reported for two species of cuttlefish (*A. esculentum* and *A. lycidas*) and three loliginid squids (*Gao and Natsukari, 1990*); 1n=36 has been reported for *A. arabica* (*Jazayeri et al., 2011*) and 1n=24 in *A. pharaonis* (*Ebrahimi Pour, 2009*). In *S. officinalis*, a karyotype of 1n=52 is reported for testis samples (*Vitturi et al., 1982*). Combining cytogenetic preparations with fluorescent labeling of centromeric or telomeric sequences, as demonstrated in the octopus *A.*

*aerolatus* (*Adachi et al., 2014*) could help resolve these issues. Establishing a routine staining protocol would enable comprehensive tests at the species- and population-level.

Taken together, our results illustrate the difficulty of assembling large genomes with high repeat content and large karyotypes, at least from sequencing data alone. Internal validation methods and genome comparisons across species are, therefore, important. Convergence of reliable estimates will, in turn, help identify chromosomal fusion-with-mixing events (FWM; fusion of two ancestral chromosomes followed by extensive shuffling of their gene content) that are clade-specific. Early branching order in Decapodiformes has been notoriously unstable (*Chen et al., 2025*; *Sanchez et al., 2026*; *López-Córdova et al., 2022*; *Tanner et al., 2017*; *Anderson and Lindgren, 2021*; *Lindgren et al., 2012*; *Sanchez et al., 2018*); thus, such rare and irreversible FWM characters could be useful in further phylogenetic analysis of this clade (*Simakov et al., 2022*; *Schultz et al., 2023*).

In addition to studying chromosomal topology in phylogenetic reconstructions, some of the most interesting aspects of these rearrangements relate to changes of and innovation in regulatory elements that underlie phenotypic diversity. In coleoid cephalopods, it is thought that an ancient large-scale genome rearrangement was combined with lineage-specific changes and repeat expansions (*Belcaid et al., 2019*; *Albertin et al., 2022*; *Marino et al., 2022*). This restructuring gave rise to hundreds of tightly linked, evolutionarily unique microsyntenies, corresponding to distinct topological compartments with specialized regulatory architectures that contribute to complex, tissue-specific expression patterns in the nervous system and elsewhere (*Schmidbaur et al., 2022*). Extending this, chromosomal conformation analyses in *E. scolopes* revealed that co-regulated eye and light-organ genes cluster at topologically associating domain (TAD) boundaries, and that an evolutionarily recent rearrangement at the dachshund (DAC) locus may have been instrumental in the emergence of the symbiotic light organ in *Euprymna* - directly linking specific chromosomal topology to morphological innovation (*Rouressol et al., 2023*).

To understand the broader functional impact of these changes across coleoids, a recent study investigating Micro-C, RNA-seq, and ATAC-seq data from multiple species revealed broadly conserved chromatin domains, but also many lineage-specific chromatin loops that form novel regulatory signatures and impact expression profiles across species and tissues (*Rogers et al., 2025*).

Despite the observed small-scale regulatory changes, the chromosomes of decapods are considered to be more closely related to the ancestral coleoid karyotype than those of octopods. The derived octopod karyotype becomes apparent when comparing it to the genome of the vampire squid, an early-branching octopodiform (sister to all octopods) which retained features of the decapod, ancestral karyotype (*Yoshida et al., 2025*). Taken together, the conserved karyotype of decapods accommodates fine-scale regulatory diversity that might underlie morphological diversity among species, which suggests that many regulatory innovations are still being evolutionarily explored through rearrangements within the existing chromosomes.

## Genome size and repeat landscape

The size of the genome of *S. officinalis* is estimated to be 5.68 Gb - comparable to those of other Decapodiformes and roughly twice the size of typical Octopodiformes genomes. We found that over 71% of the genome is repetitive, the repeats being dominated by unclassified elements and retrotransposons. The near absence of SINEs reinforces the idea that their expansion in octopuses was a lineage-specific event (*Marino et al., 2022*). These results underscore the evolutionary complexity and dynamism of cephalopod genomes, shaped by waves of transposon activity that likely played a role in the evolution of novel traits (*Schmidbaur et al., 2022*; *Albertin et al., 2022*).

## Gene annotation and comparative assessment

We annotated 18,663 gene models and 23,768 proteins using a combination of short- and long-read RNA-seq data and molluscan protein references. Compared with other annotated cuttlefish genomes (*A. lycidas*, *A. pharaonis*), our annotation is conservative concerning gene numbers but stands out in terms of completeness and taxonomic consistency. OMArk-based evaluations confirmed that our gene models have low levels of fragmentation and contamination, and high lineage-specific coherence. Striking differences across species in the number of predicted genes, duplications, and missing orthologs likely reflect variation in the annotation pipelines and reference data rather than true biological differences. Note that the comprehensive annotation of protein isoforms remains challenging

even for model organisms (*Amaral et al., 2023*; *Frankish et al., 2023*). Still, our dataset provides a solid foundation that may be refined with additional (long-read) transcriptomic data taken from diverse tissues and developmental stages.

We characterized the *S. officinalis* gene models further by clustering them into families together with 12 other cephalopod and non-cephalopod mollusks. As in other coleoid genomes, we observed large-scale expansions of gene families, such as C2H2 zinc finger transcription factors and proto-cadherins, genes implicated in regulatory innovation, neural development, and plasticity (*Albertin et al., 2015*; *Belcaid et al., 2019*; *Albertin et al., 2022*; *Styfhals et al., 2019*). These conserved expansions may be linked to the unusual cognitive complexity and behavioral sophistication of these invertebrates.

We profiled the expression of expanded gene families in bulk RNA-seq data from multiple *S. officinalis* tissues and found that many genes were differentially expressed, frequently in a tissue-restricted manner. This spatial partitioning of paralogs across tissues validates many expanded families as likely functional and biologically relevant, and suggests that subfunctionalization or neofunctionalization may have accompanied gene family expansion in *S. officinalis*, with individual family members acquiring specialized roles (*Lynch and Force, 2000*; *Force et al., 1999*). Note that we only investigated the expression of the largest copy number families, and the most significantly expanded families. Whether this pattern generalizes to the broader set of expanded families remains to be determined.

Among the large families, the cadherin and C2H2 zinc finger families showed preferential expression in neural tissues, consistent with known roles in synaptic organization and transcriptional regulation of neural development, respectively. The skin-specific expression of the RING-type zinc finger family is notable given its restricted expansion in decapods. RING domains are characteristic of E3 ubiquitin ligases that catalyze protein ubiquitination and target substrates for proteasomal degradation (*Freemont, 2000*; *Gamsjaeger et al., 2007*). Their expression could hint at unique demands in protein turnover in the skin, but the specific function of this gene family in cuttlefish skin remains to be determined.

Many of the smaller but significantly expanded gene families, which were predominantly found in decapod genomes, were linked to retrotransposons, like retrovirus-related Pol polyprotein, reverse transcriptase domain, and ribonuclease H-like families. We confirmed that these genes were genuine coding sequences and not artifacts of TE repeats, and that they were expressed in *S. officinalis* tissues. The combination of coding-sequence annotation, expression, and tissue specificity is consistent with at least some of these loci having been retained as functional genes after retroelement-related origins.

Notably, we observed many expanded families containing CCHC-type zinc fingers, which contain a 'zinc knuckle' motif, characteristic of retroviral nucleocapsid proteins (*Summers et al., 1992*). In addition to the retroviral function, some eukaryotic proteins containing CCHC-domains have been described, which likely originate from domesticated nucleocapsid sequences and play important roles in RNA metabolism (*Aceituno-Valenzuela et al., 2020*; *Wang et al., 2021*). Taken together, these gene families may represent a recent wave of retroelement activity, and the variation in copy number across different Decapodiformes is consistent with repeated recruitment and/or independent retention of retroelement-derived sequences in different decapod lineages.

Two non-retroviral gene families were also differentially expressed in *S. officinalis* tissues: a UDP-glucuronosyltransferase (UGT) family and a homeodomain-like family. The expression of the UGT family, particularly in skin and retina is interesting: enzymes of the UGT family are involved in pigment metabolism in insects (*Ahn and Marygold, 2021*), but have also been reported in mollusks as playing roles in environmental chemical tolerance (*Qi et al., 2026*). The homeodomain-like family contains transcription factors with important roles in body patterning and brain regionalization (*Focareta et al., 2014*). We found members of this family to be expressed in the vertical and subvertical lobes of the brain, but also more weakly in other brain areas, suggesting novel regulatory roles of this transcription factor class in higher-order neural circuits in brain areas associated with learning and memory in cephalopods (*Boycott and Young, 1955*).

A potential caveat of this analysis is that the gene modeling approaches used for the species were different: while the majority of genomes were annotated with the RefSeq pipeline (*Goldfarb et al., 2025*), others used Ensembl genebuild (*Aken et al., 2016*), BRAKER (*Simão et al., 2015*; *Hoff et al., 2019*; *Brůna et al., 2021*; *Brůna et al., 2024a*; *Gabriel et al., 2021*; *Hoff et al., 2016*; *Stanke et al., 2006*; *Stanke et al., 2008*; *Li, 2023*; *Iwata and Gotoh, 2012*; *Gotoh, 2008*; *Buchfink et al., 2015*;

*Kovaka et al., 2019*; *Brůna et al., 2024b*; *Huang and Li, 2023*; *Pertea and Pertea, 2020*; *Gabriel et al., 2024*), or other custom approaches. This may have introduced artificially inflated gene counts due to insufficient isoform resolution, or missing gene models due to the absence of reference data. In the future, as genome annotations become progressively refined (as for *D. pealeii Veenstra, 2025*), gene family-level analyses will become more powerful and enable the identification of additional lineage- or species-specific innovations.

## Conclusions and outlook

This study provides a new genomic resource for *Sepia officinalis*. The chromosome-scale assembly and annotation open the door to in-depth studies of gene regulation, neural circuit development, and genome evolution across coleoid cephalopods.

As we move toward a more complete picture of cephalopod genome evolution, the integration of chromosomal synteny, regulatory architecture, and transcriptomic diversity across species will be especially important. The *S. officinalis* genome represents an important step on this path, enabling high-resolution comparative and functional analyses of one of the most enigmatic and evolutionarily successful invertebrate lineages.

# Materials and methods

## Animal husbandry

All research and animal care procedures were carried out following the institutional guidelines that comply with national and international laws and policies (DIRECTIVE 2010/63/EU; German animal welfare act; FELASA guidelines). European cuttlefish *Sepia officinalis* were hatched from eggs collected in the Portuguese Atlantic and reared in a seawater system, at 20 °C. The closed system contains 4000 l of artificial seawater (ASW; Instant Ocean) with a salinity of 33 per thousand and pH of 8–8.5. Water quality was tested daily and adjusted as required. Trace elements and amino acids were supplied weekly. Marine LED lights above each tank provided a 12/12 hr light/dark cycle with gradual on- and off-sets at 07:00 and 19:00. The animals were fed live food (either *Hemimysis* spp. or small *Palaemonetes* spp.) ad libitum twice per day. The animals were housed together in 120 L glass tanks with a constant water through-flow resulting in five complete water exchanges per hour. Enrichment consisted of natural fine-grained sand substrate, seaweed (*Caulerpa prolifera*), rocks of different sizes and various natural and man-made three-dimensional objects.

## Tissue preparation

Animals underwent terminal anesthesia to prevent animal suffering or distress, following the Guidelines for the Care and Welfare of Cephalopods in Research (*Ponte et al., 2023*; *Fiorito et al., 2015*). Animals were transferred to a bucket containing ethanol 2% (v/v) in ASW. After 5 min, animals were gently probed for simple avoidance reflexes, and the ethanol concentration in the ASW was gradually increased to a maximum of 5%. Sufficient depth of anesthesia was reached when the animal no longer reacted even to stronger stimuli (touching and pinching with tweezers), and no reaction was observed when a hand was moved in the visual area and the cornea was touched (*Andrews et al., 2013*). In this deep anesthesia, the animals were decapitated and the tissue was rapidly transferred into ice-cold, calcium-free Ringer solution (460 mM NaCl, 10 mM KCl, 51 mM MgCl$_2$, 10 mM glucose, 2 mM Glutamine, 10 mM HEPES, pH 7.4) bubbled with oxygen. Brain and body tissue was dissected under a stereoscope and flash-frozen in liquid nitrogen before storage at –80 °C until further use.

Unless stated otherwise, sequencing libraries were prepared from the same individual (6-month-old adult *Sepia officinalis*, F1 from eggs collected in Portugal).

## Long-read whole genome library preparation and sequencing

The long-read library was prepared and sequenced at the MPI for Plant Research in Cologne, Germany. Genomic DNA was extracted from flash-frozen brain tissue with the MagAttract HMW DNA kit (Qiagen, Cat. no. 67563) and the sequencing library was prepared with the SMRTbell express template prep kit 2.0 (PacBio, Cat. no. 100-938-900). The library was sequenced on 5 SMRT cells on the PacBio Sequel II with the Sequel II binding kit 2.2 (102-089-000), Sequel II sequencing kit 2.0 (101-820-200), and SMRT Cell 8 M tray (101-389-001).

## Long-read RNA library preparation and sequencing

RNA was isolated from various flash-frozen tissues (different brain areas, mantle/epidermis, arm/tentacle; 5–10 mg each). First, tissue samples were homogenized in TRIzol using a homogenizer (VDI12/S12N-5S, VWR, Germany). RNA was isolated and DNAseI treated according to the Direct-zol RNA Mini prep kit from ZymoResearch (R2050). RNA was quantified by spectrophotometry (Nanodrop, Thermo Scientific, USA) and its quality was assessed with a RNA 6000 Nanochip on Bioanalyzer (Agilent Technologies, Germany).

The Iso-Seq libraries were prepared and sequenced at the Sequencing facility of the MPI for Plant Research, using a method targeting the 5' cap and poly-A tail of mRNAs, described in detail in *Cartolano et al., 2016*. Briefly, mRNA was pooled from the 8 tissue samples and cDNA was synthesized using the TeloPrime Full-Length cDNA Amplification Kit V2 (Lexogen, Cat Nr: 013.08 or 013.24). The sequencing library was prepared with the SMRTbell express template prep kit 2.0 (PacBio, Cat. no. 100-938-900). The pooled library was sequenced on 1 SMRT cell on the PacBio Sequel II system using the Sequel II binding kit 2.1 (101-843-000), Sequel II sequencing kit 2.0 (101-820-200), and SMRT Cell 8 M tray (101-389-001).

Furthermore, a separate library was prepared from optic lobe cDNA and sequenced on a second SMRT cell using the same reagents described above.

## Omni-C library preparation and sequencing

An Omni-C library was prepared and sequenced at the MPI for Plant Research in Cologne, Germany. The library was prepared from brain tissue using the Dovetail Omni-C Kit (Dovetail Genomics, Cat. No. 21005) according to the manufacturer's protocol. Briefly, chromatin was fixed in place in the nucleus. Fixed chromatin was digested with DNaseI then extracted. Chromatin ends were repaired and ligated to a biotinylated bridge adapter followed by proximity ligation of adapter-containing ends. After proximity ligation, crosslinks were reversed and the DNA was purified from proteins. Purified DNA was treated to remove biotin that was not internal to ligated fragments. The sequencing library was generated using Illumina-compatible adapters. Biotin-containing fragments were isolated using streptavidin beads before PCR enrichment of the library. The library was sequenced on an Illumina NextSeq 2000 platform to generate 400 million 2×150 bp read pairs.

## Short-read DNA sequencing

High molecular weight genomic DNA was isolated from brain tissue (optic lobe) using NEB Monarch gDNA Purification Kit (T3010S). First, 10 mg of flash-frozen tissue were submerged in 100 µl of tissue lysis buffer, cut into small pieces, and transferred into a 1.5 ml tube containing an additional 100 µl of lysis buffer. Then, 3 µl of 20 U/µl Proteinase K was added to the tissue sample and incubated for 2 hr at 56 °C in a thermal mixer with agitation until tissue pieces were completely dissolved. After incubation, tissue lysate was purified from the debris by centrifugation for 3 min at maximal speed (Eppendorf, Germany). Clean supernatant was transferred to a new 1.5 ml tube and incubated with 3 µl RNase A for 5 min at 56 °C with agitation at full speed. After RNase A incubation tissue lysate was column purified and eluted with 80 µl of elution buffer according to the manufacturer protocol. The purity of gDNA was assessed using A260/280 and A260/230 absorbance ratios measured by NanoDrop spectrophotometer (Thermo Fisher Scientific). The integrity of the obtained gDNA was checked using 0,75% agarose gel electrophoresis by loading 100 ng of gDNA sample and using HindIII digested Lambda DNA (NEB#3012) as a marker.

DNA sequencing libraries were prepared from 500 ng of high MW gDNA using Illumina DNA PCR-Free Tagmentation Library Prep Kit (20041794) and UD Indexes Set A (20026121). Obtained dual-indexed single-stranded libraries were quantified using the Qubit single-stranded DNA (ssDNA) assay kit (Q10212, Thermo Fisher Scientific). 1000 pM of the final library were run on P3 reagents and Illumina NextSeq2000 using Illumina DNA PCR-Free Sequencing and Indexing primer (20041797) with 151 bp paired-end reads.

## Short-read RNA library preparation and sequencing

For short-read RNA sequencing, tissue from another animal (8-month-old adult, F0 from eggs collected in Normandie, France) was used. RNA was isolated from various flash-frozen tissues (different brain areas, skin, and retina; 5 mg each). First, tissue samples were homogenized in TRIzol using a

homogenizer (VDI12/S12N-5S, VWR, Germany) following DNAseI treatment and column purification according to the Direct-zol RNA Micro prep kit from ZymoResearch (R2062). The RNA integrity and quantity were measured with the Qubit fluorometer (Invitrogen, Q33216) and the 2100 Bioanalyzer (Agilent Technologies, Germany).

RNA-seq libraries were prepared from 300 ng of total RNA, using the Illumina TruSeq stranded mRNA library prep kit (Cat:20020594) and IDT for Illumina xGen UDI-UMI Adapters (Cat:10005903). Libraries were sequenced on an Illumina NextSeq500 (Mid output, 300 cyc) and NextSeq2000 (P3, 300 cyc) with 145 bp paired-end reads.

## Nuclear genome assembly

Before assembly, PacBio HiFi reads were subjected to additional trimming to remove residual adapter sequences using NCBI's VecScreen tool. The k-mer distribution was estimated using Meryl (*Rhie et al., 2020*) within the Merfin (*Formenti et al., 2022*) package with a k-mer size of 21, and genome size was estimated using GenomeScope (*Ranallo-Benavidez et al., 2020*) from Illumina short reads and PacBio HiFi data. The primary assembly was generated with hifiasm (*Cheng et al., 2021*) using a combination of HiFi and Hi-C reads. Scaffolding was performed iteratively with YAHS (*Zhou et al., 2023*) on the phased haplotype 1. We adjusted the run parameters for scaffolding resolutions (`-r 1000,2000,5000,10000,20000,30000,50000,70000,100000,150000,200000,250000,300000,350000,400000,500000,700000,1000000,2000000,5000000,10000000,15000000,20000000,30000000,40000000,50000000,60000000,70000000,80000000,100000000,110000000,120000000,150000000,170000000,200000000,500000000`), repetitions per resolution (-R3), minimum read mapping quality (-q1) and telomeric sequence (--telo-motif 'TTAGGG') from the only experimentially determined telomere motif for cephalopods (*Amphioctopus areolatus Adachi et al., 2014*, accessed via TeloBase *Lyčka et al., 2024*) to optimize the tool's performance for a large, fragmented assembly. Finally, the assembly was manually curated using JBAT (*Dudchenko et al., 2018*) to place residual scaffolds into chromosome scaffolds. Different versions of the assembly were evaluated based on Hi-C coverage, mRNA alignment, and completeness based on BUSCO (*Simão et al., 2015*) v5.5.0 (*metazoa_odb10* and *mollusca_odb10*).

## Mitochondrial genome assembly

To assemble the mitochondrial genome, we aligned a published *S. officinalis* mitochondrial genome (*Akasaki et al., 2006*) (NC_007895.1) to the PacBio Hi-Fi and Omni-C reads using minimap2 (*Li, 2018*). The hits were extracted using seqtk (*Li, 2013*) subseq and assembled using hifiasm (*Cheng et al., 2021*). The resulting contigs contained multiple repeats of the circular mitochondrial genome, and were aligned back to NC_007895.1 to extract the final sequence.

## Nuclear genome annotation

Repetitive elements in the genome were softmasked using RepeatMasker v4.1.7-p1 (*Smit et al., 2025*) (with rmblast v2.14.1+) (-xsmall and -gff options) and after creating a custom repeat library with RepeatModeler v.2.0.6 (*Flynn et al., 2020*) (without LTRstruct option).

Gene models were created using BRAKER3 (*Simão et al., 2015*; *Hoff et al., 2019*; *Brůna et al., 2021*; *Brůna et al., 2024a*; *Gabriel et al., 2021*; *Hoff et al., 2016*; *Stanke et al., 2006*; *Stanke et al., 2008*; *Li, 2023*; *Iwata and Gotoh, 2012*; *Gotoh, 2008*; *Buchfink et al., 2015*; *Kovaka et al., 2019*; *Brůna et al., 2024b*; *Huang and Li, 2023*; *Pertea and Pertea, 2020*; *Gabriel et al., 2024*), run via the Docker container, on the softmasked genome using both RNA-seq and protein data. We used long-read Iso-Seq data and short-read Illumina RNA-seq data from various tissues (brain, skin, mantle, retina) generated for this study (see above). For protein data, publicly available proteomes for *Doryteuthis pealeii* (*Albertin et al., 2022*), *Euprymna scolopes* (*Rogers et al., 2024*), *Octopus bimaculoides* (*Albertin et al., 2022*), *Octopus vulgaris* (*Destanović et al., 2023*), *Nautilus pompilius* (*Zhang et al., 2021*), *Pecten maximus* (*Zeng et al., 2021*) were used for training, as well as a previously generated proteome for *Sepia officinalis* from StringTie gene models (this study, described below). RNA-seq data was input into BRAKER3 using the `--bam` option, protein files were specified with the `--prot_seq` option. Untranslated regions (UTRs) were added by the -addUTR=on parameter. The BUSCO completeness of the resulting gene set was maximized using TSEBRA within BRAKER on the BUSCO lineage *metazoa_odb10*.

For the initial protein set used as input for BRAKER, the short and long RNA reads were aligned to the genome using *minimap2* (*Li, 2018*). Transcript models were predicted with StringTie v3.0.0 (*Shumate et al., 2022*) using the `--conservative` and `--mix` options. The resulting GTF files were combined using the transcript merge mode, resulting in a set of non-redundant transcripts. Finally, TransDecoder v5.7.0 (*Haas, 2026*) was run with default parameters to translate coding regions in the transcripts.

The final gene annotations for *S. officinalis* were assessed for completeness using OMArk v.0.3.0 (*Nevers et al., 2022*) with the ancestral clade Lophotrochozoa and without, including splice information, accessed via their webserver. Proteins were annotated with orthology information using InterProScan (*Blum et al., 2025*) v5.73–104, including lookup of annotations and GO terms (options -iprlookup -goterms). Further orthology information was added using the eggNOG-mapper v2.1.12 (*Cantalapiedra et al., 2021*) webserver with the eggNOG v5.0 database and default parameters.

## Whole genome alignment and synteny analysis

For whole genome alignments, the assembly produced for *Sepia officinalis* in this study and the published assembly from the Darwin Tree of Life project (GCA_964300435.1) or the *Acanthosepion esculentum* genome (GCA_964036315.1) were aligned using Winnowmap2 (*Jain et al., 2022*) and visualized with a custom script in R v4.4.2 (*R Development Core Team, 2024*).

For synteny analyses, proteome and gtf files were downloaded for *Doryteuthis pealeii* and *Euprymna scolopes*. Annotation files for *Acanthosepion esculentum* were recently generated by lift-over annotation from *Acanthosepion pharaonis* (*Coffing et al., 2025*) and downloaded from Zenodo (*Coffing et al., 2024*).

Synteny analyses between all chromosomes of the compared species were performed using the R package GENESPACE v.1.2.3 (*Lovell et al., 2022*) with default parameters, described briefly below. Protein sequence similarity was first estimated using DIAMOND2 (*Buchfink et al., 2015*) in fast mode, and orthogroups and pairwise orthologues were inferred using OrthoFinder v2.5 (*Emms et al., 2025*) with hierarchical orthogroups (HOGs) enabled. Prior to synteny inference, tandem arrays were condensed to their most central representative gene, and gene rank order was recalculated on these array-representative genes to reduce confounding effects of tandem duplication on collinearity detection.

Syntenic blocks were identified pairwise between all genome combinations using MCScanX (*Wang et al., 2012*), constrained to DIAMOND hits where both query and target genes belonged to the same orthogroup (onlyOgAnchors = TRUE). Initial anchor hits were clustered into large syntenic regions using a density-based spatial clustering approach (dbscan *Hahsler et al., 2019*), with a minimum block size of five anchor genes (blkSize = 5) and a maximum of five intervening non-anchor genes permitted within a block (nGaps = 5). Anchor clustering used a search radius of 25 gene-rank positions (blkRadius = 25). All hits falling within a syntenic buffer of 100 gene-rank positions around confirmed block anchors (synBuff = 100) were retained as syntenic. No secondary syntenic hits were included (nSecondaryHits = 0). Syntenic orthogroups were integrated across all pairwise comparisons and collapsed into a pan-genome annotation anchored to *S. officinalis* as the reference genome.

Syntenic relationships were visualized as riparian plots and pairwise dotplots using the built-in plotting functions of GENESPACE v1.2.3. Riparian plots were constructed using physical chromosomal coordinates (useOrder = FALSE) with *S. officinalis* as the reference, displaying all three genomes. A second riparian plot was generated highlighting a region of interest. Pairwise dotplots were produced for the *S. officinalis–D. pealeii* and *S. officinalis–E. scolopes* genome comparisons, displaying only synteny-validated hits (type = 'syntenic') with a minimum synteny score of 10 (minScore = 10) and a minimum of 10 genes per chromosome pair required for display (minGenes2plot = 10).

## Coverage analysis

For the analysis of breakpoints between *S. officinalis* assemblies, Hi-C data was aligned using bwa-mem2 v2.3 (*Vasimuddin et al., 2019*) and quantified using pairtools v1.1.0 (*Abdennur et al., 2023*). Contacts between scaffolds (trans pairs) were extracted from deduplicated read pairs (pair type UU). For each of the four breakpoints, all trans pairs between the two flanking scaffold halves were extracted using awk. As controls, the corresponding joined scaffold from the opposing assembly

was paired with the nearest-length uninvolved scaffold from the same assembly, and trans pairs were extracted identically.

Genome-wide background and intra-scaffold contact distributions were computed in a single pass over each pairs file using awk, recording the total number of trans pairs and intra-scaffold pairs (>1 kb separation) for all scaffold pairs across the genome. Trans contact rates were normalized by the product of the two scaffold lengths (pairs per Mb²) to correct for length bias. For visualization, genome-wide trans pair positions were binned into 500 kb windows along each scaffold to produce contact density tracks. To test whether trans contact rates at breakpoints were elevated above background, empirical p-values were computed as the fraction of all genome-wide scaffold pair rates equal to or exceeding the observed breakpoint rate. For the three DToL breakpoints jointly, a one-tailed Wilcoxon rank-sum test was applied against the background distribution.

HiFi reads were aligned to the scaffolded assemblies using minimap2 (*Li, 2018*) and duplicate-marked alignments were removed. HiFi read depth was computed over the terminal 200 kb at each scaffold end using pysam v0.22.1 (*pysam-developers, 2026*) count_coverage with a mapping quality threshold of ≥10, binned at 1 kb resolution. Spanning reads - defined as reads with supplementary alignments crossing a scaffold boundary - were identified by querying split alignments at each junction using pysam.

Repeat content at scaffold junctions was characterized by extracting 200 kb windows flanking each breakpoint and the corresponding correct scaffold termini, and running RepeatMasker v4.1.7-p1 (*Smit et al., 2025*) (with rmblast v2.14.1+) against a de novo repeat library generated by RepeatModeler v.2.0.6 (*Flynn et al., 2020*) from the same assembly (described above). Repeat annotations were parsed and collapsed into eight classes (SINE, LINE, DNA transposons, LTR elements, simple repeats, low-complexity regions, unknown repeats, and other), and the masked fraction per class was quantified for each window.

For sex chromosome analysis, read coverage across chromosomes was analyzed as described recently (*Coffing et al., 2025*). Short reads were aligned using STAR v2.7.11b (*Dobin et al., 2013*) to our chromosome-scale assembly. For *A. esculentum*, short reads (ERR12945500) and assembly (GCA_964036315.1) were downloaded from NCBI. The sequencing depth was calculated using mosdepth (*Pedersen and Quinlan, 2018*) by a window size of 500,000 bp and normalized to the median coverage of the first chromosome. Chromosomes with significantly reduced read coverage were identified by a one-sided Wilcoxon rank-sum test of each chromosome's normalized depth windows against all remaining chromosomes. P-values were corrected using the Benjamini-Hochberg method and considered only for chromosomes with at least 10% decrease in median normalized depth.

## Sex genotyping by qPCR

The sex of the F1 individual used for whole-genome sequencing was determined by qPCR quantification of the relative copy number of an autosomal locus and a sex-chromosomal locus, adapted from the protocol published recently (*Rubino et al., 2025*). Primers targeting an autosomal locus on chromosome 2 (SepOff_chr2_auto_G2_F, 5′-TTTGCCACTGTGTCCCTTTATAC-3′; SepOff_chr2_auto_G2_R, 5′-ACACACACAGGCTGCTTATTG-3′) and a sex-chromosomal locus on chromosome 46 (SepOff_chr46_sex_H2_F, 5′-TTTCAACCCATCTGCGTCTATAG-3′; SepOff_chr46_sex_H2_R, 5′-ACTCCTCTCGTTGCATGATTAC-3′) were designed by *Rubino et al., 2025* and validated against the *S. officinalis* assembly generated in this study; chromosome 46 was identified as the *S. officinalis* sex chromosome on the basis of chromosome syteny with *A. esculentum*. Primers were synthesized by Integrated DNA Technologies (IDT; 100 nmol scale, standard desalting).

Genomic DNA isolated from optic lobe tissue using the NEB Monarch Spin gDNA Extraction Kit (T3010S, as described above for short-read DNA sequencing) was used as input. qPCR reactions were assembled in 96-well Bio-Rad Hard-Shell PCR plates with KAPA SYBR FAST qPCR Master Mix (2×Universal; Roche/KAPA Biosystems) and 0.5 µM of each primer, sealed with Bio-Rad Microseal 'B' film (MSB1001), and amplified on a Bio-Rad CFX96 Real-Time PCR Detection System with the following program: initial denaturation at 95 °C for 2 min; 45 cycles of 95 °C for 15 s and 60 °C for 45 s; followed by a melt curve from 60°C to 95°C (ramp rate 0.06 °C/s). Four technical replicates were performed per primer pair. Mean quantification cycle (Cq) values were calculated across the four replicates after inspecting melt curves to confirm single-product amplification; the Cq difference ΔCq

= A_Cq − S_Cq was used to classify the individual, with positive values diagnostic of males (ZZ) and negative values diagnostic of females (Z0). The F1 individual was classified as male.

## Gene family expansion analysis

Orthogroups were inferred across 13 molluscan species (*Table 2*), including *S. officinalis*, using OrthoFinder v3.1.0 (*Emms et al., 2025*) with default parameters. The input proteomes included the longest protein isoform per gene for each species. The rooted species tree from OrthoFinder (*Emms and Kelly, 2018*; *Emms and Kelly, 2017*) was converted to an ultrametric tree using the R package ape (*Paradis and Schliep, 2019*) v5.8.1.

Gene families were filtered by removing orthogroups present in only a single species, and by separating orthogroups containing 100 or more gene copies in any species, as extreme copy-number differences in gene families prevent likelihood calculation under the applied birth-death model.

Gene family evolution rates were estimated using CAFE5 (*Mendes et al., 2021*) v5.1.1 on the filtered orthogroups, using the ultrametric species tree as input. Four models were evaluated: the base model (single global lambda), and Gamma models with k=2, 3, and 4 rate categories, which allow evolutionary rate variation among gene families. The Gamma k=3 model was selected based on the best (lowest) final log-likelihood score. All subsequent statistical inferences were performed under this model.

For families showing statistically significant expansion or contraction (*p*<0.05 after Bonferroni correction), branch-specific copy-number changes were extracted from the CAFE5 output. Families were categorized as *S. officinalis*-specific, coleoid-specific, or broad expansions based on the distribution of significant changes across the phylogeny.

To assess whether expanded gene families in *S. officinalis* contained genes derived from or embedded within repetitive elements, a coordinate-based overlap analysis was performed. For each gene in an expanded orthogroup, the overlap between its coding sequence (CDS) coordinates and RepeatMasker annotations was computed using *bedtools intersect* v2.30 (*Quinlan and Hall, 2010*). To avoid double-counting when multiple repeat annotations overlapped the same coding bases, overlapping repeat intervals were merged per gene prior to summing covered bases, and the overlap fraction was computed as merged covered bases divided by total CDS length.

## Bulk RNA-seq analysis

Quality-filtered paired-end RNA-seq reads were aligned to the *S. officinalis* genome assembly using STAR (*Dobin et al., 2013*) (v2.7.11b) with the following parameters: `--outSAMmultNmax` 1 (retaining only uniquely mapping reads), `--outFilterIntronMotifs` RemoveNoncanonical (removing reads with non-canonical splice junctions), and `--outSAMtype` BAM SortedByCoordinate.

Gene-level read counts were quantified using featureCounts (*Liao et al., 2014*) from the Subread package (v2.0.8) with the gene annotation described above (GTF format). Counting was performed at the exon level (-t exon) and summarized by gene (-g gene_id), using paired-end mode (-p `--countReadPairs`) to count fragments rather than individual reads. Only reads with mapping quality ≥255 (-Q 255), corresponding to uniquely mapped reads in the STAR output, were included in the count matrix. The resulting gene count matrix was used as input for differential expression (DE) analysis.

Tissue-specific marker genes were identified using DESeq2 (*Love et al., 2014*) (v1.42.0) with a one-vs-all comparison strategy. For each tissue, samples were grouped into a binary condition ('target' vs. 'other'), where 'target' represented the tissue of interest and 'other' comprised all remaining tissues. Raw count matrices were filtered to retain genes with ≥10 counts in at least three samples prior to analysis. DE testing was performed using the Wald test, log2 fold changes were shrunk using the apeglm (*Zhu et al., 2019*) method. Genes were classified as tissue markers if they met the following criteria: adjusted p-value (Benjamini-Hochberg FDR)<0.05, log2 fold change >1, and mean expression in the target tissue >5.

To generate gene-level GO (*Aleksander et al., 2026*; *Ashburner et al., 2000*) annotations for the *S. officinalis* genome, GO term assignments from the InterProScan output (detailed earlier) were used. First, transcript IDs were collapsed to gene IDs and for genes with multiple transcripts, GO terms were aggregated using a union strategy, collecting all unique GO terms across all isoforms and annotation sources to maximize coverage. GO terms present in fewer than 5 genes or more than 500 genes in the

expressed gene universe were excluded from enrichment testing to reduce noise from overly specific or generic terms.

GO enrichment analysis was performed using the clusterProfiler (*Yu et al., 2012*; *Xu et al., 2024*) (v4.12.6) enricher function with custom GO annotations generated as described above. For each gene set (DE genes for each tissue), enrichment was tested against the background universe of all expressed genes (genes passing the low-count filter in DESeq2). Over-representation was assessed using the hypergeometric test, with p-values adjusted for multiple testing using the Benjamini-Hochberg procedure. GO terms with adjusted p-value (FDR)<0.05 were considered significantly enriched.

*S. officinalis* members of each expanded gene family were cross-referenced against tissue-specific marker genes identified by DESeq2 (as described earlier). For families with at least one differentially expressed member, GO enrichment was performed using enricher() from clusterProfiler as described for tissue marker genes. For families with at least one enriched GO term, expression patterns of differentially expressed members were visualized as a heatmap of z-scored VST values, with samples grouped by tissue.

## Acknowledgements

We thank Bruno Huettel for PacBio and Omni-C sequencing. We thank Xitong Liang, Theodosia Woo, and Mathieu Renard for help with tissue dissection. We thank Darrin Schultz, Dalila Destanović, and Thea Rogers for helpful discussions and advice on manual curation and gene modeling. We thank Victor Nieto Caballero for initial discussions on gene family expansion analysis. This work was funded by the Max Planck Society (GL) and the European Research Council (GL; ERC grant CAMOUFLAGE, 101141501). OS was supported by the ERC's Horizon 2020: European Union Research and Innovation Programme, grant No. 945026. All authors declare no conflict of interest. Views and opinions expressed are those of the authors only and do not necessarily reflect those of the European Union or the European Research Council Executive Agency. Neither the European Union nor the granting authority can be held responsible for them.

## Additional information

### Funding

| Funder | Grant reference number | Author |
|---|---|---|
| European Research Council | Advanced Grant CAMOUFLAGE 101141501 | Gilles Laurent |
| European Research Council | Horizon 2020 European Union Research and Innovation Programme grant No. 945026 | Oleg Simakov |

The funders had no role in study design, data collection and interpretation, or the decision to submit the work for publication. Open access funding provided by Max Planck Society.

### Author contributions

Simone Daniela Rencken, Conceptualization, Resources, Data curation, Formal analysis, Validation, Investigation, Visualization, Writing – original draft, Project administration, Writing – review and editing; Georgi Tushev, Data curation, Software, Formal analysis, Validation, Visualization, Writing – review and editing; David Hain, Resources, Writing – review and editing; Elena Ciirdaeva, Resources, Investigation, Writing – review and editing; Oleg Simakov, Conceptualization, Formal analysis, Supervision, Validation, Writing – original draft, Project administration, Writing – review and editing; Gilles Laurent, Conceptualization, Supervision, Funding acquisition, Writing – original draft, Project administration, Writing – review and editing

### Author ORCIDs

Simone Daniela Rencken (iD) https://orcid.org/0009-0003-9341-2898
Georgi Tushev (iD) https://orcid.org/0000-0002-3340-9422

David Hain https://orcid.org/0000-0002-8979-7938
Gilles Laurent https://orcid.org/0000-0002-2296-114X

### Ethics

All research and animal care procedures were carried out following the institutional guidelines that comply with national and international laws and policies (DIRECTIVE 2010/63/EU; German animal welfare act; FELASA guidelines). Animals underwent terminal anesthesia for tissue collection to prevent animal suffering or distress, following the Guidelines for the Care and Welfare of Cephalopods in Research.

Reviewer #1 (Public review): https://doi.org/10.7554/eLife.107393.3.sa1
Reviewer #2 (Public review): https://doi.org/10.7554/eLife.107393.3.sa2
Reviewer #3 (Public review): https://doi.org/10.7554/eLife.107393.3.sa3
Author response https://doi.org/10.7554/eLife.107393.3.sa4

---

## Additional files

### Supplementary files

MDAR checklist

### Data availability

The genome assembly and raw data can be found at the BioProject PRJNA1091451 on NCBI. Raw sequencing reads are deposited at SRA (study accession SRP570862). The code for the genome assembly and annotation is available at https://gitlab.mpcdf.mpg.de/mpibr/laur/cuttlefishomics/soffgenome (copy archived at *Tushev, 2026*). Genome annotation files are deposited at https://doi.org/10.17617/3.CGO7QG.

The following datasets were generated:

| Author(s) | Year | Dataset title | Dataset URL | Database and Identifier |
| --- | --- | --- | --- | --- |
| Rencken S, Tushev G, Hain D, Ciirdaeva E, Simakov O, Laurent G | 2025 | Sepia officinalis isolate:GLC-03058 (common cuttlefish) | https://www.ncbi.nlm.nih.gov/bioproject/PRJNA1091451 | NCBI BioProject, PRJNA1091451 |
| Rencken S, Tushev G, Hain D, Ciirdaeva E, Simakov O, Laurent G | 2025 | Chromosome-scale genome assembly of the European common cuttlefish Sepia officinalis | https://doi.org/10.17617/3.CGO7QG | Edmond - the Open Research Data Repository of the Max Planck Society, 10.17617/3.CGO7QG |

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

# Appendix 1

## Appendix 1—key resources table

| Reagent type (species) or resource | Designation | Source or reference | Identifiers | Additional information |
|---|---|---|---|---|
| Biological sample (*Sepia officinalis*) | European cuttlefish, F1 individual | Eggs supplied by Flying Sharks – consultoria e inovação, Lda., Horta, Azores, Portugal | | 6-month-old adult; F1 from eggs collected in the Portuguese Atlantic; used for long-read DNA, Iso-Seq, Omni-C and short-read DNA library preparation |
| Biological sample (*Sepia officinalis*) | European cuttlefish, F0 individual | Eggs supplied by Université de Caen Normandie, France | | 8-month-old adult; F0 from eggs collected in Normandie, France; used for short-read RNA-seq library preparation |
| Commercial assay or kit | MagAttract HMW DNA Kit | Qiagen | Cat#67563 | Genomic DNA extraction from flash-frozen brain tissue for PacBio HiFi library |
| Commercial assay or kit | NEB Monarch gDNA Purification Kit | New England Biolabs | Cat#T3010S | Genomic DNA extraction from optic lobe; used both for short-read Illumina library preparation and as template for sex-genotyping qPCR |
| Commercial assay or kit | Direct-zol RNA Miniprep Kit | Zymo Research | Cat#R2050 | RNA isolation and DNase I treatment for Iso-Seq libraries |
| Commercial assay or kit | Direct-zol RNA Microprep Kit | Zymo Research | Cat#R2062 | RNA isolation and DNase I treatment for short-read RNA-seq libraries |
| Commercial assay or kit | TeloPrime Full-Length cDNA Amplification Kit V2 | Lexogen | Cat#013.08; Cat#013.24 | Full-length cDNA synthesis targeting 5' cap and poly-A tail for Iso-Seq |
| Commercial assay or kit | SMRTbell express template prep kit 2.0 | PacBio | Cat#100-938-900 | Long-read library preparation for both HiFi DNA and Iso-Seq sequencing |
| Commercial assay or kit | Sequel II binding kit 2.2 | PacBio | Cat#102-089-000 | Used for HiFi DNA sequencing |
| Commercial assay or kit | Sequel II binding kit 2.1 | PacBio | Cat#101-843-000 | Used for Iso-Seq sequencing |
| Commercial assay or kit | Sequel II sequencing kit 2.0 | PacBio | Cat#101-820-200 | Used for PacBio Sequel II runs (HiFi DNA and Iso-Seq) |
| Commercial assay or kit | SMRT Cell 8 M tray | PacBio | Cat#101-389-001 | 5 SMRT cells used for HiFi DNA sequencing; 2 SMRT cells for Iso-Seq (pooled tissues +optic lobe) |
| Commercial assay or kit | Dovetail Omni-C Kit | Dovetail Genomics | Cat#21005 | Omni-C proximity ligation library prepared from brain tissue |
| Commercial assay or kit | Illumina DNA PCR-Free Tagmentation Library Prep Kit | Illumina | Cat#20041794 | Short-read DNA library preparation from 500 ng of high-MW gDNA |
| Commercial assay or kit | IDT for Illumina DNA/RNA UD Indexes, Set A | Illumina | Cat#20026121 | Dual indexes for short-read DNA library |
| Commercial assay or kit | Illumina DNA PCR-Free Sequencing and Indexing primer | Illumina | Cat#20041797 | Used during NextSeq2000 P3 sequencing of short-read DNA library |
| Commercial assay or kit | Qubit ssDNA Assay Kit | Thermo Fisher Scientific | Cat#Q10212 | Quantification of dual-indexed single-stranded short-read DNA libraries |
| Commercial assay or kit | Illumina TruSeq Stranded mRNA Library Prep Kit | Illumina | Cat#20020594 | Short-read RNA-seq libraries prepared from 300 ng total RNA |
| Commercial assay or kit | IDT for Illumina xGen UDI-UMI Adapters | Integrated DNA Technologies (IDT) | Cat#10005903 | Adapters used with TruSeq Stranded mRNA library prep |
| Commercial assay or kit | Illumina NextSeq500 mid output flow cell (300 cycles) | Illumina | Cat#20024905 | Used for short-read RNA sequencing |
| Commercial assay or kit | Illumina NextSeq2000 P3 flow cell (300 cycles) | Illumina | Cat#20040561 | Used for short-read RNA and DNA sequencing |
| Commercial assay or kit | KAPA SYBR FAST qPCR Master Mix (2×Universal) | Roche / KAPA Biosystems | Cat#KK4600 | qPCR master mix used for sex-chromosome genotyping qPCR |
| Chemical compound, drug | TRIzol Reagent | Invitrogen / Thermo Fisher Scientific | Cat#15596026 | Homogenization reagent for RNA isolation from flash-frozen tissues (Iso-Seq and short-read RNA-seq) |

*Appendix 1 Continued on next page*

*Appendix 1 Continued*

| Reagent type (species) or resource | Designation | Source or reference | Identifiers | Additional information |
|---|---|---|---|---|
| Sequence-based reagent | SepOff_chr2_auto_G2_F (qPCR primer) | *Rubino et al., 2025*; 10.1101/2025.10.28.685099 | | 5'-TTTGCCACTGTGTCCCTTTATAC-3'; forward primer targeting an autosomal locus on chromosome 2; used in qPCR sex genotyping; synthesized from IDT at 100 nmol DNA-oligo scale with standard desalting |
| Sequence-based reagent | SepOff_chr2_auto_G2_R (qPCR primer) | *Rubino et al., 2025*; 10.1101/2025.10.28.685099 | | 5'-ACACACACAGGCTGCTTATTG-3'; reverse primer targeting an autosomal locus on chromosome 2; used in qPCR sex genotyping; synthesized from IDT at 100 nmol DNA-oligo scale with standard desalting |
| Sequence-based reagent | SepOff_chr46_sex_H2_F (qPCR primer) | *Rubino et al., 2025*; 10.1101/2025.10.28.685099 | | 5'-TTTCAACCCATCTGCGTCTATAG-3'; forward primer targeting a sex-chromosomal locus on chromosome 46 used in qPCR sex genotyping; synthesized from IDT at 100 nmol DNA-oligo scale with standard desalting |
| Sequence-based reagent | SepOff_chr46_sex_H2_R (qPCR primer) | *Rubino et al., 2025*; 10.1101/2025.10.28.685099 | | 5'-ACTCCTCTCGTTGCATGATTAC-3'; reverse primer targeting a sex-chromosomal locus on chromosome 46 used in qPCR sex genotyping; synthesized from IDT at 100 nmol DNA-oligo scale with standard desalting |
| Other | Lambda DNA-HindIII Digest | New England Biolabs | Cat#3012 | Molecular weight ladder; 100 ng loaded alongside gDNA on 0.75% agarose gel to assess DNA integrity |
| Other | Hard-Shell 96-Well PCR Plates | Bio-Rad | Cat#HSP9601 | 96-well qPCR plates used for sex-chromosome genotyping |
| Other | Microseal 'B' PCR Plate Sealing Film | Bio-Rad | Cat#MSB1001 | Adhesive sealing film used to seal 96-well qPCR plates for sex-chromosome genotyping |
| Software, algorithm | VecScreen | NCBI; https://www.ncbi.nlm.nih.gov/tools/vecscreen/ | RRID:SCR_016577 | Adapter/vector trimming of PacBio HiFi reads prior to assembly |
| Software, algorithm | Meryl | *Rhie et al., 2020*; 10.1186/s13059-020-02134-9 | | k-mer counting (k=21) for k-mer distribution estimation; bundled with Merqury |
| Software, algorithm | Merfin | *Formenti et al., 2022*; 10.1038/s41592-022-01445-y | | Provides Meryl wrapper used for k-mer distribution estimation |
| Software, algorithm | GenomeScope 2.0 | *Ranallo-Benavidez et al., 2020*; 10.1038/s41467-020-14998-3 | RRID:SCR_017014 | Genome size estimation from Illumina short reads and PacBio HiFi data |
| Software, algorithm | hifiasm | *Cheng et al., 2021*; 10.1038/s41592-020-01056-5 | RRID:SCR_021069 | Primary genome assembly from combined HiFi+Hi C reads; also used for mitochondrial assembly |
| Software, algorithm | YAHS | *Zhou et al., 2023*; 10.1093/bioinformatics/btac808 | RRID:SCR_022965 | Hi-C scaffolding on phased haplotype 1 with custom -r/-R/-q/--telo-motif parameters |
| Software, algorithm | JBAT (Juicebox Assembly Tools) | *Dudchenko et al., 2018*; 10.1101/254797 | | Manual curation of scaffolds into chromosome-scale scaffolds |
| Software, algorithm | BUSCO v5.5.0 | *Simão et al., 2015*; 10.1093/bioinformatics/btv351 | RRID:SCR_015008 | Assembly and annotation completeness assessment using metazoa_odb10, metazoa_ob12, mollusca_odb10 and mollusca_odb12 lineages |
| Software, algorithm | minimap2 | *Li, 2018*; 10.1093/bioinformatics/bty191 | RRID:SCR_018550 | Used for: aligning mt genome reference NC_007895.1 to long reads; aligning short and long RNA reads to genome; aligning HiFi reads to scaffolded assemblies for coverage |
| Software, algorithm | seqtk | *Li, 2013* | RRID:SCR_018927 | seqtk subseq used to extract reads matching mt genome reference for mitochondrial assembly |
| Software, algorithm | RepeatMasker v4.1.7-p1 | *Smit et al., 2025*; http://www.repeatmasker.org | RRID:SCR_012954 | Soft-masking of repetitive elements (-xsmall, -gff); also used to characterize repeat content at scaffold junctions; run with rmblast v2.14.1+ |
| Software, algorithm | RepeatModeler v2.0.6 | *Flynn et al., 2020*; 10.1073/pnas.1921046117 | RRID:SCR_015027 | De novo repeat library construction (without LTRstruct option) |

*Appendix 1 Continued on next page*

*Appendix 1 Continued*

| Reagent type (species) or resource | Designation | Source or reference | Identifiers | Additional information |
|---|---|---|---|---|
| Software, algorithm | BRAKER3 (incl. TSEBRA) | *Simão et al., 2015*; *Hoff et al., 2019*; *Brůna et al., 2021*; *Gabriel et al., 2021*; *Hoff et al., 2016*; *Stanke et al., 2006*; *Stanke et al., 2008*; *Li, 2023*; *Iwata and Gotoh, 2012*; *Gotoh, 2008*; *Buchfink et al., 2015*; *Kovaka et al., 2019*; ; *Huang and Li, 2023*; *Pertea and Pertea, 2020*; *Gabriel et al., 2024*; 10.1007/978-1-4939-9173-0_5 | RRID:SCR_018964 | Gene model prediction via Docker container on softmasked genome; used both RNA-seq (--bam) and protein (--prot_seq) input; UTRs added with --addUTR=on; TSEBRA tuned to maximize BUSCO completeness on metazoa_odb10 |
| Software, algorithm | StringTie v3.0.0 | *Shumate et al., 2022*; 10.1371/journal.pcbi.1009730 | RRID:SCR_016323 | Transcript model prediction with --conservative and --mix options; GTFs merged with transcript merge mode |
| Software, algorithm | TransDecoder v5.7.0 | *Haas, 2026*; https://github.com/TransDecoder/TransDecoder | RRID:SCR_017647 | Translation of coding regions in transcripts (default parameters) |
| Software, algorithm | OMArk v0.3.0 | *Nevers et al., 2022*; 10.1101/2022.11.25.517970 | | Annotation completeness assessment; ancestral clade Lophotrochozoa; run on webserver without splice information |
| Software, algorithm | InterProScan v5.73–104 | *Blum et al., 2025*; 10.1093/nar/gkae1082 | RRID:SCR_005829 | Protein orthology and GO annotation with options -iprlookup -goterms |
| Software, algorithm | eggNOG-mapper v2.1.12 | *Cantalapiedra et al., 2021*; 10.1093/molbev/msab293 | RRID:SCR_021165 | Functional/orthology annotation via webserver with eggNOG v5.0 database, default parameters |
| Software, algorithm | Winnowmap2 | *Jain et al., 2022*; 10.1038/s41592-022-01457-8 | RRID:SCR_025349 | Whole-genome pairwise alignments of *S. officinalis* and *A. esculentum* (GCA_964036315.1) assemblies |
| Software, algorithm | R v4.4.2 | *R Development Core Team, 2024* | RRID:SCR_001905 | Statistical environment for downstream analyses and visualization (whole-genome alignment plots and other custom scripts) |
| Software, algorithm | GENESPACE v1.2.3 | *Lovell et al., 2022*; 10.7554/eLife.78526 | | Pairwise synteny analysis across all chromosomes of compared species with default parameters; riparian plots and pairwise dotplots |
| Software, algorithm | DIAMOND2 | *Buchfink et al., 2015*; 10.1038/nmeth.3176 | RRID:SCR_016071 | Protein sequence similarity in fast mode within GENESPACE |
| Software, algorithm | OrthoFinder v2.5 | *Emms and Kelly, 2019*; 10.1186/s13059-019-1832-y | RRID:SCR_017118 | Orthogroup and pairwise orthologue inference with hierarchical orthogroups (HOGs); used within GENESPACE |
| Software, algorithm | OrthoFinder v3.1.0 | *Emms et al., 2025*; 10.1101/2025.07.15.664860 | RRID:SCR_017118 | Orthogroup inference across 13 molluscan species for gene family expansion analysis; default parameters; rooted species tree generated via STAG (*Ponte et al., 2023*) and STRIDE (*Andrews et al., 2013*) |
| Software, algorithm | MCScanX | *Wang et al., 2012*; 10.1093/nar/gkr1293 | RRID:SCR_022067 | Pairwise syntenic block identification (onlyOgAnchors = TRUE, blkSize = 5, nGaps = 5, blkRadius = 25, synBuff = 100, nSecondaryHits = 0) |
| Software, algorithm | dbscan (R package) | *Hahsler et al., 2019*; 10.18637/jss.v091.i01 | | Density-based clustering of MCScanX anchor hits into syntenic regions |
| Software, algorithm | bwa-mem2 v2.3 | *Vasimuddin et al., 2019*; 10.1109/IPDPS.2019.00041 | RRID:SCR_022192 | Alignment of Hi-C reads for breakpoint coverage analysis |
| Software, algorithm | pairtools v1.1.0 | *Abdennur et al., 2023*; 10.1101/2023.02.13.528389 | RRID:SCR_023038 | Quantification of Hi-C contacts; extraction of trans pairs from deduplicated read pairs (pair type UU) |
| Software, algorithm | pysam v0.22.1 | *pysam-developers, 2026*; https://github.com/pysam-developers/pysam | RRID:SCR_021017 | HiFi read depth via count_coverage (MAPQ ≥10, 1 kb bins); spanning reads identified by querying split alignments |
| Software, algorithm | STAR v2.7.11b | *Dobin et al., 2013*; 10.1093/bioinformatics/bts635 | RRID:SCR_004463 | Alignment of short reads to chromosome-scale assembly (sex chromosome analysis and RNA-seq) |
| Software, algorithm | mosdepth | *Pedersen and Quinlan, 2018*; 10.1093/bioinformatics/btx699 | RRID:SCR_018929 | Sequencing coverage calculation for sex chromosome analysis |

*Appendix 1 Continued on next page*

*Appendix 1 Continued*

| Reagent type (species) or resource | Designation | Source or reference | Identifiers | Additional information |
|---|---|---|---|---|
| Software, algorithm | ape v5.8.1 (R package) | *Paradis and Schliep, 2019*; 10.1093/bioinformatics/bty633 | RRID:SCR_017343 | Conversion of rooted OrthoFinder species tree to ultrametric tree |
| Software, algorithm | CAFE5 v5.1.1 | *Mendes et al., 2021*; 10.1093/bioinformatics/btaa1022 | RRID:SCR_018924 | Gene family evolution rate estimation |
| Software, algorithm | bedtools v2.30 | *Quinlan and Hall, 2010*; 10.1093/bioinformatics/btq033 | RRID:SCR_006646 | bedtools intersect for CDS–RepeatMasker overlap analysis of expanded gene family members |
| Software, algorithm | featureCounts (Subread v2.0.8) | *Liao et al., 2014*; 10.1093/bioinformatics/btt656 | RRID:SCR_012919 | Gene-level read counting from STAR-aligned RNA-seq BAMs (-t exon, -g gene_id, -p --countReadPairs, -Q 255) |
| Software, algorithm | DESeq2 v1.42.0 | *Love et al., 2014*; 10.1186/s13059-014-0550-8 | RRID:SCR_015687 | Tissue marker identification in bulk RNA-seq data |
| Software, algorithm | apeglm | *Zhu et al., 2019*; 10.1093/bioinformatics/bty895 | RRID:SCR_026951 | log2 fold-change shrinkage applied to DESeq2 results |
| Software, algorithm | clusterProfiler v4.12.6 | *Yu et al., 2012*; 10.1089/omi.2011.0118 | RRID:SCR_016884 | GO enrichment via enricher() with custom GO annotations from InterProScan |

