## [Editor Report · eLife Assessment]

This manuscript reports a high-quality genome assembly of the European cuttlefish, *Sepia officinalis*, a representative species of the Cephalopod lineage. This **solid** work relies on current best practices in genome sequencing and assembly, combining PacBio HiFi long reads and Hi-C chromatin conformation capture, and on state-of-the-art comparative genomic analyses, including chromosome number evolution and analyses of expanded gene families. The resulting genome will be a **valuable** resource for researchers interested in cuttlefish biology and comparative genomics in general.

---

## [Referee Report · Reviewer #1 (Public review)]

[Editors' note: this version has been assessed by the Reviewing Editor without further input from the original reviewers. The authors have carefully considered all the reviewers' comments. The newly added analyses, figures, and text sections are of high quality, and we commend the authors for their in-depth revision of the manuscript.]

This manuscript presents a high-quality, chromosome-level genome assembly of the European cuttlefish (*Sepia officinalis*), a representative species of the cephalopod lineage. Using state-of-the-art sequencing and scaffolding technologies -including PacBio HiFi long reads and Hi-C chromatin conformation capture - the authors deliver a genome assembly with exceptional contiguity and completeness, as evidenced by high BUSCO scores. This genome resource fills a significant gap in cephalopod genomics and offers a valuable foundation for studies in neurobiology, behavior, and evolutionary biology. However, there are several major aspects that need to be strengthened.

---

## [Referee Report · Reviewer #2 (Public review)]

This paper concerns an interesting organism, *Sepia officinalis*. However, in the opinion of this reviewer, the paper reads somewhat like a genome report. The authors have used 23x PacBio HiFi in conjunction with relatively low coverage (11x) Hi-C to scaffold the genome into a karyotype of 47 chromosomes. They have used a combination of short and long read RNA seq to annotate the genome in what looks like a very good annotation. The paper offers basic analyses of the Busco evaluation, some descriptive analyses of gene family and repeat content, and a bit more focused analysis on synteny among sequenced squids. Generally, the data will be useful.

---

## [Referee Report · Reviewer #3 (Public review)]

Summary:

In this study, authors Simone Rencken and co-authors present and investigate the genome of the common cuttlefish *Sepia officinalis*.

Strengths:

The authors explain in a detailed yet concise manner the main steps for a genome assembly, with very robust methods for validation, and according to current best practices. In addition to the chromosomal assembly, the authors confirmed the presence of 47 chromosomes using Hi-C data and multiple species synteny. They also generated a comprehensive gene annotation, with assessments of gene completeness, providing a useful resource for the community of researchers interested in cuttlefish biology and comparative genomics.

---

## [Author Response]

The following is the authors’ response to the original reviews.

**Public Reviews:**

**Reviewer 1 (Public review):**
Summary:This manuscript presents a high-quality, chromosome-level genome assembly of the European cuttlefish (*Sepia officinalis*), a representative species of the cephalopod lineage. Using state-of-the-art sequencing and scaffolding technologies -including PacBio HiFi long reads and Hi-C chromatin conformation capture - the authors deliver a genome assembly with exceptional contiguity and completeness, as evidenced by high BUSCO scores. This genome resource fills a significant gap in cephalopod genomics and offers a valuable foundation for studies in neurobiology, behavior, and evolutionary biology. However, there are several major aspects that need to be strengthened.Major Revisions Recommended:(1) Single-individual genome limitationThe genome assembly is based on a single individual, which appears to be male. While this approach is common in genome projects, it does not capture the full genetic diversity of the species. As S. officinalis exhibits a wide geographical range and possible population structure, future efforts (or discussion in this manuscript) should consider re-sequencing multiple individuals - of both sexes and from diverse geographic origins - to characterize population-level variation, sex-linked features, and structural polymorphisms.

We thank the reviewer for this summary and the important point raised. While sequencing additional individuals, unfortunately, lies outside the scope of our study, we used the published data from the DToL assembly (from a male individual from a different geographical origin) to begin to investigate their differences.

First, we attempted to create a mixed assembly from both datasets, as also suggested by Reviewer 2, to increase data coverage and genetic information. Even though the heterozygosity estimate is quite low (ca. 1%), the mixed assembly produced severely inflated and fragmented results, yielding an assembly ca. 3× larger than expected, with the top 46 contigs covering only ~5% of the total length - a sign of over duplication and failed haplotype collapse.

This result is not surprising when considering the assembly algorithms: most programs, including hifiasm used in this study, assume a single diploid individual (or a trio assembly including data from both parents), so using multiple individuals breaks this assumption. Assembly pipelines infer homozygous/heterozygous coverage cutoffs from the k-mer histogram. Mixing individuals raises apparent heterozygosity far above true diploid levels, turning the expected bimodal k-mer profile into a complex multimodal distribution. This misleads the phasing and purging steps in the assembly pipeline, causing over-expansion and fragmentation of the assembly.

Second, we created separate assemblies from the raw data sets of MPIBR and DToL using the exact same pipeline and parameters to avoid the technical problem described above. These assemblies are directly comparable, and after aligning them, it is possible to build a pangenome graph that we believe would help to address the points raised by the reviewer. Pangenome graphs can represent cross-individual variation more accurately and improve read alignment in regions of high genomic variation, which can aid population-level analyses [1]. We agree on the importance of this work, yet collecting data from more individuals and the construction and analysis of a pangenome graph lies beyond the scope of this manuscript and should be part of future efforts by the cephalopod genomics field.

(2) Limited experimental validation of chromosomal inferencesThe study reports chromosome-scale scaffolding using Hi-C data and proposes a revised karyotype for S. officinalis. However, these inferences would be significantly strengthened by orthogonal validation methods. In particular, fluorescence in situ hybridization (FISH) or karyotyping from cytogenetic preparations would provide direct confirmation of chromosome number and structural arrangements. The reliance solely on Hi-C contact maps for inferring chromosomal organization should be acknowledged as a limitation or supplemented with such validations.

We appreciate the reviewer’s point regarding the value of orthogonal validation methods to support the chromosome-scale scaffolding and proposed karyotype. We acknowledge that relying solely on Hi-C contact maps to infer chromosome number and structure presents limitations, as also becomes apparent in our detailed analysis of both *S. officinalis* genome assemblies (in Figure 2 and Supplementary Figure 3 of the revised manuscript). We attempted to complement these analyses with cytogenetic approaches. Unfortunately, the availability of suitable mitotic tissue was limited. Moreover, our karyotyping trials proved challenging: resolving the ≥92 (2n) chromosomes in situ was not feasible due to their high number and the small size of the nuclei (approximately 5 µm in diameter on average).

We now highlight this point as an important direction for future work in our discussion (line 456-466):

“Additional methods such as cytogenetic karyotyping or optical mapping such as BioNano [141] (imaging of fluorescently tagged, linearized DNA) could be used to validate chromosome numbers. However, whereas karyotypes of octopuses have been consistent throughout the literature (1n=30) [142,143], those measured in decapods vary greatly. For example, 1n=46 chromosomes have been reported for two species of cuttlefish (*A. esculentum* and *A. lycidas*) and three loliginid squids [85]; 1n=36 has been reported for *A. Arabica* [86] and 1n=24 in *A. pharaonis* [87]. In *S. officinalis*, a karyotype of 1n=52 is reported for testis samples [88]. Combining cytogenetic preparations with fluorescent labeling of centromeric or telomeric sequences, as demonstrated in the octopus *A. aerolatus* [143] could help resolve these issues. Establishing a routine staining protocol would enable comprehensive tests at the species- and population-level.”

(3) Shallow discussion of chromosomal evolutionThe manuscript briefly mentions chromosomal number differences among cephalopods but does not explore their evolutionary or functional implications. A more thorough comparative analysis - linking chromosomal rearrangements (e.g., fusions, fissions) with ecological adaptation, life history, or neural complexity - would greatly enhance the impact of the findings. Referencing chromosomal dynamics in related taxa and possible links to behavioral innovations would contextualize these results more effectively.

We agree with the reviewer that this is a fascinating topic of research that demands further attention and have extended our discussion, which now reads (line 476-501):

“In addition to studying chromosomal topology in phylogenetic reconstructions, some of the most interesting aspects of these rearrangements relate to changes of and innovation in regulatory elements that underlie phenotypic diversity. In coleoid cephalopods, it is thought that an ancient large-scale genome rearrangement was combined with lineage-specific changes and repeat expansions [48–50]. This restructuring gave rise to hundreds of tightly linked, evolutionarily unique microsyntenies, corresponding to distinct topological compartments with specialized regulatory architectures that contribute to complex, tissue-specific expression patterns in the nervous system and elsewhere [43]. Extending this, chromosomal conformation analyses in *E. scolopes* revealed that co-regulated eye and light-organ genes cluster at topologically associating domain (TAD) boundaries, and that an evolutionarily recent rearrangement at the dachshund (DAC) locus may have been instrumental in the emergence of the symbiotic light organ in *Euprymna* - directly linking specific chromosomal topology to morphological innovation [44].

To understand the broader functional impact of these changes across coleoids, a recent study investigating Micro-C, RNA-seq, and ATAC-seq data from multiple species revealed broadly conserved chromatin domains, but also many lineage-specific chromatin loops that form novel regulatory signatures and impact expression profiles across species and tissues [149].

Despite the observed small-scale regulatory changes, the chromosomes of decapods are considered to be more closely related to the ancestral coleoid karyotype than those of octopods. The derived octopod karyotype becomes apparent when comparing it to the genome of the vampire squid, an early-branching octopodiform (sister to all octopods) which retained features of the decapod, ancestral karyotype [150]. Taken together, the conserved karyotype of decapods accommodates fine-scale regulatory diversity that might underlie morphological diversity among species, which suggests that many regulatory innovations are still being evolutionarily explored through rearrangements within the existing chromosomes.”

(4) Underdeveloped gene family and pathway analysisWhile the authors identify expansions in gene families such as protocadherins and C2H2 zinc finger transcription factors, the functional significance of these expansions remains speculative. The manuscript would benefit from:(a) Functional enrichment analyses (e.g., GO, KEGG) targeting these gene families.(b) Expression profiling across tissues or developmental stages to infer regulatory roles.(c) Comparison with expression or expansion patterns in other cephalopods with known behavioral complexity (e.g., Octopus bimaculoides, Euprymna scolopes).(d) Potential integration of transcriptomic or epigenomic data to support regulatory hypotheses.

We thank the reviewer for these constructive suggestions and have substantially expanded the functional characterization of expanded gene families in the revised manuscript.

To address points (a) + (b), we performed GO enrichment analyses for all expanded gene families (orthogroups), both for the largest gene families and the most significantly expanded families identified from our CAFE5 analysis. Further, we cross-referenced all *S. officinalis* members of each expanded orthogroup against differentially expressed genes in our bulk RNA-seq data from multiple tissues (initially collected to improve the gene modeling), allowing us to infer tissue-specific expression patterns for the expanded families.

To address point (c), the species-resolved copy-number profiles from our orthogroup analysis directly situate the *S. officinalis* expansions within the broader coleoid context, including *O. bimaculoides*, *O. vulgaris*, *E. scolopes*, and *D. pealeii*, enabling direct comparison of expansion scale and lineage specificity across species with varying degrees of behavioural complexity. We note that the C2H2 zinc finger and protocadherin expansions show distinct phylogenetic profiles consistent with independent radiations in octopods and decapodiforms, in agreement with recent studies.

Regarding point (d), no epigenomic data for *S. officinalis* was publicly available at the time of writing, thus we focused on the transcriptomic data from this study, as described above.

We describe this analysis in two additional results paragraphs to the manuscript, one modified (Figure 4) and two new figures (Figure 5 and Supplementary Figure 7), which are reproduced (lines 294-400):

“Analysis of expanded gene families

We sought to investigate the *S. officinalis* gene annotation and place it in the context of gene repertoires from other cephalopod or molluscan species. First, we collected available genome annotations from 12 other molluscan species (Table 2) and clustered them using OrthoFinder v.3.1.0 [122], resulting in 23,658 orthogroups, hereafter named gene families.

First, we investigated 36 of the gene families that contain more than 100 genes in any of the species, with 17 of these families containing at least one gene of *S. officinalis*, that reflect large-scale gene family expansions (Figure 4E). We used the InterProScan and eggNOG-mapper annotations to infer functional roles of these genes, selecting the most common gene annotation as the name of the gene family.

The zinc finger C2H2-type transcription factors (TFs) were grouped into three of the large gene families, with the largest family (OG0000000) only present in decapod cephalopods. This likely reflects the largely independent expansions in the octopod and decapod lineages that date back to a burst of transposon activity ca. 25 million years ago [46,48,49]. The largest expansion across mollusks occurs in the cadherin-like family (OG0000001): 310 in *S. officinalis*, 283 in *D. pealeii*, 209 in *A. lycidas*, 102 in *O. vulgaris*, 55 in *O. bimaculoides*, with low but non-zero counts in bivalves (C. virginica, *M. gigas*). This profile is consistent with the protocadherin expansion first described in *O. bimaculoides* [46] and subsequently shown to be present across cephalopods [48,49,123].

HPGDS (OG0000005, hematopoietic prostaglandin D synthase) is a glutathione-S-transferase family member that catalyzes the conversion of prostaglandins, which have well-described roles in immune responses in vertebrates and insects [124,125]. This family shows a broad expansion in decapods, with a lesser expansion in octopods. Additionally, members of the glutathione-S-transferase families have been co-opted as S-crystallins, structural proteins found in the lens of cephalopods that may, or may not, retain enzymatic functions [126,127].

Two large families are mostly lineage-restricted. The RING-type zinc finger family (OG0000058) has 103 copies in *S. officinalis* and 26 in *A. lycidas* but is absent in all other species except for *E. scolopes*. Conversely, OG0000002 (unknown function) has 479 copies in *E. scolopes* and only a few copies in the other species. This interesting Sepiolid-specific expansion warrants further characterization.

We estimated gene family evolution rates using CAFE5 [128] for all families with less than 100 copies in any species (this excludes the families described above, as very large copy-number differences between species preclude likelihood calculations under the applied birth-death model). After comparing different model parameters, we chose a gamma model with three rate categories, allowing for evolutionary rate variation among gene families. Out of the 12,895 gene families analyzed, 1,813 showed a significant (p < 0.05) expansion or contraction in at least one of the species. We focused our analysis on the 30 most significantly expanded families; among them were several retrotransposon-associated domains that have expanded specifically in *S. officinalis* five families carrying Retrovirus-related Pol polyprotein domains, two Reverse transcriptase domain families, and four Ribonuclease H-like families (Supplementary Figure 7A). There was no coordinate-based overlap of the coding sequences with annotated TEs from the RepeatMasker output (Methods).

In addition to the three large gene families of C2H2 zinc finger expansions, 45 gene families containing this TF type showed a significant change in the CAFE5 analysis. Notably, eight of the significant gene families, as well as four of the largest gene families, were annotated as CCHC-type zinc fingers, which contain a “zinc knuckle” motif that is characteristic of retroviral nucleocapsid proteins [129] and is functionally integrated in the genomes of several species, including humans [130].

Some gene families without any relationship to retrotransposons were also expanded. For example, the UGT2A1-related family is a UDP-glucuronosyltransferase, a class of enzymes central to phase II detoxification and conjugation of metabolites, reported in other mollusks in the context of environmental chemical tolerance [131], and in insects in the context of pigmentation [132]. We also detected a family of homeodomain-like proteins, representing an expansion of this important TF family.

Tissue-specific expression of expanded gene families

To place the identified gene families in a functional context, we profiled their expression in the bulk RNA-seq data (taken from multiple tissues of *S. officinalis*) used originally for gene modeling (Figure 5A). Principal component analysis (PCA) revealed the largest axis of variation in gene expression to separate brain tissues from peripheral tissues, with skin being the most transcriptomically distinct (Figure 5A), consistent with the high number of tissue-specific differentially expressed (DE) genes identified in non-neural tissues (Figure 5B). We identified the genes belonging to expanded families that were differentially expressed across tissues and enriched gene ontology [133,134] (GO) terms for them to gain additional insight. The large families excluded from CAFE5 modelling and the significantly expanded families identified by CAFE5 were analyzed separately.

Eleven of the largest gene families were expressed in our data (Figure 5C) and five had enriched GO terms (Figure 5D,E). Among them, the cadherin family showed brain-restricted expression and GO terms related to cell–cell adhesion and calcium binding, consistent with their role in neuronal connectivity and circuit formation [46,135]. Two C2H2 zinc finger gene families were expressed in the optic and vertical/subvertical lobes of the brain and in the skin, with GO terms related to DNA-binding, transcriptional regulation or development. The RING-type zinc finger family was expressed specifically in the skin, with GO terms including zinc binding and ubiquitin protein ligase activity, the canonical function of RING-domain E3 ligases [136]. Genes of the HPGDS/S-crystallin family were expressed in the brain (basal and optic lobes and posterior subesophageal mass) and skin, with GO terms related to glutathione metabolism, matching their described enzymatic function. We did not find expression in the retina, which is expected given that S-crystallins are expressed in lentigenic cells of the eye [42,137] and these cells were not included during sampling.

Among the 30 most significantly expanded families examined (out of 1,813 total), expression was widespread (20/30) and tissue-specific differential expression was common (17/30), suggesting that a substantial proportion of expanded paralogs represent functional coding sequences with specialized spatial deployment (Supplementary Figure 7B). Ten of the retrotransposon-associated families were differentially expressed in the brain (optic and vertical/subvertical lobes) and skin, arguing against these loci being inactive repeat fragments and supporting their inclusion as transcribed gene models. Two significantly expanded families showed both differential expression and enriched GO terms (Supplementary Figure 7C). The first was the UGT2A1-related family, which had the largest number of differentially expressed genes overall, with expression concentrated in the skin, retina and posterior subesophageal mass of the brain. Enriched GO terms matched the described enzymatic function for this family, namely UDP-glycosyltransferase activity. The second gene family was the homeodomain-like family with enrichment for DNA binding terms consistent with their role as transcription factors, and was preferentially expressed in the vertical and subvertical brain lobes with weaker expression in other areas.

Collectively, many differentially expressed genes from expanded families were restricted to specific tissues or brain subregions (Figure 5F and Supplementary Figure 7D), indicating that paralogs within an expanded family have adopted distinct spatial expression domains and possibly, specialized functions.”

**Reviewer 2 (Public review):**
Summary:This paper concerns an interesting organism, *Sepia officinalis*. However, in the opinion of this reviewer, the paper reads somewhat like a genome report. The authors have used 23x PacBio HiFi in conjunction with relatively low coverage (11x) Hi-C to scaffold the genome into a karyotype of 47 chromosomes. They have used a combination of short and long read RNA seq to annotate the genome in what looks like a very good annotation. The paper offers basic analyses of the Busco evaluation, some descriptive analyses of gene family and repeat content, and a bit more focused analysis on synteny among sequenced squids. Generally, the data will be useful.Strengths:This is a high-quality annotation, and the data ultimately will be useful to other researchers. I appreciate trying to understand what's happening between assemblies of S. officinalis.Weaknesses:I don't believe the data at hand makes a strong case for the argument of 47 chromosomes. This is my biggest sticking point with the paper, and it is for a few reasons:(1) The authors point to assembly differences between the DToL assembly and the one presented in the manuscript and seem to claim that DToL is incorrect. However, the DToL assembly (xcSepOffi3.1) is based on much deeper HiFi and HiC coverage than the one at hand (51x and 80+x respectively). There are many things to try here, including:(a) Downloading the DToL data and reassembling using a common pipeline.(b) Downsampling the DToL data to similar coverage as what the authors have achieved.(c) Combining your data and that of DToL for even deeper coverage (heterozygosity is low enough that I don't imagine this impeding things too badly).

We thank the reviewer for these helpful suggestions and want to clarify that we did not seek to point out errors in the DToL assembly, but rather to investigate the unexpected discrepancies between the two assemblies. It is correct that the DToL data has a much higher coverage than our data. We followed the individual suggestions and incorporated them into the revised manuscript. We reproduce the relevant sections below, and provide additional information:

(a) Downloading the DToL data and reassembling using a common pipeline.

We downloaded the DToL data and reassembled it using a common pipeline, yielding the results listed in Author response table 1. The DToL assembly is more contiguous, which is mainly due to its higher HiFi coverage. It also receives slightly better BUSCO scores (computed using odb12 as recommended by Reviewer 3).

**Author response table 1. sa4table1:** 

	soff250801_mpibr			soff250801_dtol		
contigs	p_ctg	hap1	hap2	P_ctg	hap1	hap2
records	8.289	10.651	10.425	8.783	11.026	11.089
length.raw	6.049.669.443	5.675.386.986	5.662.586.038	6.053.996.452	5.721.157.269	5.950.565.264
length._min	2.496	2.959	2.959	5.151	5.999	6.915
length.n25	825.171	518.304	518.787	865.207	560.043	580.263
length.n50	1.723.203	1.032.632	1.010.375	1.810 .137	1.165.578	1.182.649
length.n75	3.129.788	1.845.042	1.789 .755	3.317 .923	2.125.467	2.203.556
length.max	14.924.420	7.470 .229	10.284.785	15.032.877	9.223.473	11.008.627
length.med	309.429	293.763	314.184	245.676	239.367	250.010
length.avg	729.843	532.850	543.173	689.285	518.878	536.618
length.top46	339.372.830	213.767 .346	225.912.626	364.533 .811	278.738 .760	283.344.104
frac.top46	5,609774769	3,766568633	3,98956633	6,021374705	4,872069529	4,761633415

Full statistics of S. officinalis assemblies from two independent datasets, assembled using a common pipeline.

The updated manuscript now reads (lines 146-159):

“A chromosome-scale assembly for *Sepia officinalis* was released recently by the Wellcome Sanger Institute’s Darwin Tree of Life project [75] (DToL, GCA_964300435.1). That genome was assembled from a male individual using high coverage PacBio Sequel II (~51x) and Arima2 Hi-C (~80x) data, with a final assembly size of 5.8 Gb. The the haploid chromosome number was estimated to be 49. To compare both *S. officinalis* datasets directly, we downloaded the DToL data and created two new assemblies using the pipeline described above (hifiasm using PacBio HiFi and Hi-C data). The resulting assemblies were overall very similar, with the DToL assembly having a slightly higher contiguity (N50 length, see Table 1) and BUSCO completeness (Supplementary Figure 2A,B) due to their higher sequencing coverage.”

To further compare the two datasets, we added a new Figure 2 to the revised manuscript and the following paragraph to the results (lines 160-169):

“After scaffolding with YAHS, both datasets reached the previously identified chromosome numbers (1n=47 for MPIBR and 1n=49 for DToL, Figure 2A,B). To further investigate this surprising discrepancy, we aligned both assemblies using Winnowmap [89] to locate the differences between them (Figure 2C). We observed four “breakpoints” (BP) of chromosome scaffolds: one in the MPIBR assembly compared to DToL (BP1: DToL_5 = MPIBR_40+44) and three in the DToL assembly compared to MPIBR (BP2: DToL_31+40 = MPIBR_2, BP3: DToL_41+46 = MPIBR_6, BP4: DToL_44+45 = MPIBR_7). We also aligned the assemblies to the chromosome-scale genome of another cuttlefish *Acanthosepion esculentum* (1n=46, GCA_964036315.1). In this alignment, all four breakpoints were collinear with single *A. esculentum* chromosomes (Figure 2D).”

(b) Downsampling the DToL data to similar coverage as what the authors have achieved.

Instead of downsampling the DToL data, we decided to analyze the Hi-C and HiFi data for both assemblies, focusing on the four “breakpoints” between the assemblies and the *A. esculentum* genome that we described above. First, we performed a QC analysis of the Hi-C reads using pairtools [2], the result is visualized in Author response image 1. The percentage of valid Hi-C read pairs, i.e., cis pairs with insert distances of more than 1 kb and trans pairs, following the Dovetail genomics QC manual (here). When Hi-C pairs were aligned to the primary contigs from hifiasm (as is used for scaffolding with YAHS), the DToL HiC data contains fewer valid read pairs (11.4%) than the MPIBR data (43.1%), possibly due to using a different tissue (eye vs. optic lobe) and HiC kit (Arima 2 vs. Dovetail OmniC) for the library preparation. Nonetheless, due to the much higher overall coverage, the amount of valid read pairs is still 2.35x higher for DToL (144,014,368 pairs) than for MPIBR (61,318,955 pairs). The higher trans fraction (i.e. HiC pairs across contigs) is dependent on the length of the primary contigs, so the higher trans fraction for the MPIBR data can be explained by the lower contiguity of its primary contigs. It is conceivable that for both assemblies, the low numbers of valid read pairs introduce a technical fragmentation of certain chromosomes, as indicated by the identified breakpoints (Figure 2).

**Author response image 1. sa4fig1:** Analysis of Hi-C read pairs from both *S. officinalis* assemblies. Hi-C reads were aligned to the primary contigs from hifiasm (as is used for scaffolding with YAHS) and analyzed using pairtools. Note the higher fraction of long-range contacts (at least 1 kb cis pairs or trans pairs) in the MPIBR data (top) compared to DToL (bottom). Due to overall higher coverage, the absolute number of read pairs is higher for DToL than for MPIBR data.

Second, we performed a detailed analysis of read coverage along the breakpoint junctions of the discrepant chromosomes/scaffolds between both assemblies. We included a description of the results and a new Supplementary Figure 3 in the manuscript, (lines 171-207):

“To better understand the potential cause of these divergent chromosome numbers, we analyzed the Hi-C and HiFi coverage in the breakpoint regions (Supplementary Figure 3A). First, we aligned the Hi-Fi reads to the scaffolds and extracted all alignments along the 200 kb terminal scaffold windows to find any notable drops in coverage, or reads spanning any of the scaffold junctions. We detected no spanning reads. This is not surprising given that no contigs were assembled at these sites, resulting in the observed scaffold junctions. More interestingly, we noted a ~5-fold decrease in HiFi coverage along the DToL scaffold_40 (part of BP2) relative to its flanking regions, indicating a highly repetitive, low-mappability region at this boundary.

Next, we realigned the Hi-C data to the scaffolded assemblies using bwa-mem2 [91] and extracted all trans HiC pairs (between-scaffold contacts) using pairtools [92]. We normalized trans HiC contacts to the scaffold length and compared contact rates between breakpoint scaffolds to the baseline contact rate (computed from pairs of scaffolds with a clear 1-to-1 match between assemblies), and the contact rate within scaffolds (intra-scaffold pairs) (Supplementary Figure 3B,C). The contact rates within breakpoints were consistently lower than within scaffolds, likely falling below the threshold to be merged during assembly. However, the contact rates at three of four breakpoints (BP1, BP3, BP4) were significantly elevated above the genome-wide background distribution (empirical p = 0.010, 0.005, 0.005 respectively), suggesting that they may represent intra-chromosomal contacts disrupted by a misassembly. Notably, BP2 was not significant (empirical p = 0.170), likely due to the low coverage and mappability around the DToL scaffold_40 boundary. Considered jointly, the three DToL breakpoint scaffold pairs showed significantly higher trans contact rates than the background (Wilcoxon rank-sum, one-tailed, U = 1771, p = 0.004).

Lastly, we analyzed the repeat landscape around the 200 kb scaffold ends using RepeatMasker [93] and the custom repeat library that we had generated for *Sepia officinalis* (described further below). Compared to control scaffolds of the same assembly, we observed consistently elevated repeat content at the breakpoint junctions (mean 71.5% vs 67.6% masked bases), with an enrichment of unclassified repeats (32.1% vs 30.0%), which could explain a repeat-driven assembly fragmentation or scaffolding failure. The BP2 DToL scaffold_40 junction window was 99.99% masked (99.2% unclassified repeats), providing a likely mechanistic explanation for both the HiFi coverage drop and the absence of a significant trans Hi-C signal at this breakpoint. Taken together, these analyses suggest that the different chromosome numbers across the two *S. officinalis* assemblies are due to technical reasons, caused by repeat-rich scaffold boundaries that impair HiFi and Hi-C read alignment and in turn, correct assembly in these regions.”

(c) Combining your data and that of DToL for even deeper coverage (heterozygosity is low enough that I don't imagine this impeding things too badly).

When combining the data to achieve a higher coverage, we ran into the assembly fragmentation issues detailed above in response (1) to Reviewer 1.

(2) Looking at Figure 1, there appears to be a misjoin at chromosome 42. Looking carefully at Figure S1, that misjoin does not appear on any of the panels - this is confusing. Given the size of that chromosome and the authors' chromosome numbering, I'm guessing this is a manual merge as it's larger than most of the chromosomes numerically close (40, 41, 43, etc). Further, staring closely at Figure 1, there appear to be cross-scaffold contacts between 42 and 43 and 42 and 44. Secondarily there are contacts between 43 and 44. This bit of the assembly seems potentially problematic.

This is a great observation, indeed the HiC maps differ between Figure 1 and Figure S1. Figure 1 is the result of scaffolding with YAHS and manual curation, whereas Figure S1 was scaffolded using HapHiC. We updated the figure legend to clarify this important difference. HapHiC produces very clean contact maps without the need for manual curation, but when analyzed at a higher resolution, the tool broke many contigs and ultimately compromised the assembly quality, possibly due to our comparatively low HiC coverage. Thus, we preferred to use YAHS and manual curation, which is perhaps inherently error-prone, as becomes apparent in the regions of the assembly that are pointed out by the reviewer.

**Reviewer 3 (Public review):**
Summary:In this study, authors Simone Rencken and co-authors present and investigate the genome of the common cuttlefish *Sepia officinalis*.Strengths:The authors explain in a detailed yet concise manner the main steps for a genome assembly, with very robust methods for validation, and according to current best practices. In addition to the chromosomal assembly, the authors confirmed the presence of 47 chromosomes using Hi-C data and multiple species synteny. They also generated a comprehensive gene annotation, with assessments of gene completeness, providing a useful resource for the community of researchers interested in cuttlefish biology and comparative genomics.Weaknesses:While the study touches upon the subjects of gene content, TE activity, or species-level comparisons, the study does not provide in-depth investigations of these.

We thank the reviewer for their positive assessment of our manuscript. We acknowledge the descriptive nature and limitations of our previous analyses of gene content, TE distribution, and species comparisons. Our focus for the initial submission was to provide a high-quality assembly that could serve as a resource for anyone interested in *Sepia officinalis* or related species. However, we agree that greater insight into genome content is valuable as well. In the revised manuscript, we included a more detailed analysis of expanded gene families and GO enrichment analysis of our bulkRNAseq data, which we summarized in response (4) to reviewer 1.

**Recommendations for the authors:**

**Reviewer #1 (Recommendations for the authors):**
Minor Revisions Recommended:(1) Figure and legend claritySeveral figures lack sufficient annotation. All figures, including supplementary ones, should include:(a) Clear axis labels.(b) Descriptions of statistical measures (n values, error bars, statistical tests).(c) Legends that allow the figure to be understood independently of the main text.

We updated the figures accordingly.

(2) Terminology and formatting(a) Consistency in gene and species nomenclature should be maintained throughout (e.g., italicizing gene names and Latin binomials).(b) Ensure that abbreviations (e.g., Hi-C, BUSCO, FISH) are defined upon first use.

We updated the nomenclature throughout the text and checked the definition of abbreviations used in the text. Further, we updated the names of several cuttlefish species according to the recent revision of genera, e.g. *Sepia esculenta* was changed to *Acanthosepion esculentum* [3].

(3) Literature coverageThe references primarily focus on earlier studies from 2010-2020. It would strengthen the context to include recent high-impact studies on cephalopod genomics and chromosomal biology published in the last 3 years (e.g., 2022-2024).

We apologize for this oversight and have extended the manuscript to discuss more of these recent studies.

(4) Clarify methodsWhile the methods section is generally detailed, some critical aspects are underspecified:(a) Parameters used in genome annotation tools (e.g., BRAKER, RepeatMasker).

We thank the reviewer for bringing our attention to this shortcoming, and have added the missing parameters to the methods section. Additionally, the full code is available at here

(b) Criteria for ortholog clustering and gene family expansion analysis.

The details have been added to the methods section, which now reads (lines 828-853):

“Orthogroups were inferred across 13 molluscan species (Table 2), including *S. officinalis*, using OrthoFinder v3.1.0 [122] with default parameters. The input proteomes included the longest protein isoform per gene for each species. The rooted species tree from OrthoFinder [182,184] was converted to an ultrametric tree using the R package ape [183] v5.8.1.

Gene families were filtered by removing orthogroups present in only a single species, and by separating orthogroups containing 100 or more gene copies in any species, as extreme copy-number differences in gene families prevent likelihood calculation under the applied birth-death model.

Gene family evolution rates were estimated using CAFE5 [128] v5.1.1 on the filtered orthogroups, using the ultrametric species tree as input. Four models were evaluated: the base model (single global lambda), and Gamma models with k = 2, 3, and 4 rate categories, which allow evolutionary rate variation among gene families. The Gamma k = 3 model was selected based on the best (lowest) final log-likelihood score. All subsequent statistical inferences were performed under this model.

For families showing statistically significant expansion or contraction (p < 0.05 after Bonferroni correction), branch-specific copy-number changes were extracted from the CAFE5 output. Families were categorized as *S. officinalis*-specific, coleoid-specific, or broad expansions based on the distribution of significant changes across the phylogeny.

To assess whether expanded gene families in *S. officinalis* contained genes derived from or embedded within repetitive elements, a coordinate-based overlap analysis was performed. For each gene in an expanded orthogroup, the overlap between its coding sequence (CDS) coordinates and RepeatMasker annotations was computed using *bedtools intersect* v2.30 [185]. To avoid double-counting when multiple repeat annotations overlapped the same coding bases, overlapping repeat intervals were merged per gene prior to summing covered bases, and the overlap fraction was computed as merged covered bases divided by total CDS length.”

(c) Thresholds or cutoffs for synteny or duplication detection.

We included the details in the updated methods (lines 755-781):

“Synteny analyses between all chromosomes of the compared species were performed using the R package GENESPACE v.1.2.3 [175] with default parameters, described briefly below. Protein sequence similarity was first estimated using DIAMOND2 [109] in fast mode, and orthogroups and pairwise orthologues were inferred using OrthoFinder v2.5 [176] with hierarchical orthogroups (HOGs) enabled. Prior to synteny inference, tandem arrays were condensed to their most central representative gene, and gene rank order was recalculated on these array-representative genes to reduce confounding effects of tandem duplication on collinearity detection.

Syntenic blocks were identified pairwise between all genome combinations using MCScanX [177], constrained to DIAMOND hits where both query and target genes belonged to the same orthogroup (onlyOgAnchors = TRUE). Initial anchor hits were clustered into large syntenic regions using a density-based spatial clustering approach (dbscan [178]), with a minimum block size of five anchor genes (blkSize = 5) and a maximum of five intervening non-anchor genes permitted within a block (nGaps = 5). Anchor clustering used a search radius of 25 gene-rank positions (blkRadius = 25). All hits falling within a syntenic buffer of 100 gene-rank positions around confirmed block anchors (synBuff = 100) were retained as syntenic. No secondary syntenic hits were included (nSecondaryHits = 0). Syntenic orthogroups were integrated across all pairwise comparisons and collapsed into a pan-genome annotation anchored to. *S. officinalis* was used as the reference genome.

Syntenic relationships were visualized as riparian plots and pairwise dotplots using the built-in plotting functions of GENESPACE v1.2.3. Riparian plots were constructed using physical chromosomal coordinates (useOrder = FALSE) with *S. officinalis* as the reference, displaying all three genomes. A second riparian plot was generated highlighting a region of interest. Pairwise dotplots were produced species for the *S. officinalis*–*D. pealeii* and *S. officinalis*–*E. scolopes* genome comparisons, displaying only synteny-validated hits (type = "syntenic") with a minimum synteny score of 10 (minScore = 10) and a minimum of 10 genes per chromosome pair required for display (minGenes2plot = 10).”

**Reviewer #2 (Recommendations for the authors):**
Line 153 should be supplemental Figure 3B.

The text was referring to the correct Figure 2B (three species synteny comparison). It is now updated to Figure 3B in the revised manuscript.

**Reviewer #3 (Recommendations for the authors):**
(1) L37: Perhaps add a comparison with other species (mammals, Drosophila, etc.) to put this number in context.

We agree with this recommendation and added numbers for *Drosophila* and mouse to the text (lines 40-45):

“Coleoid cephalopods (octopus, squid, cuttlefish) are a highly derived group of mollusks, characterized by the largest nervous systems among all invertebrates (ca. 500 million neurons in an adult octopus of which 200 million are in the central brain [1,2], compared to ca. 140,000 in the fruit fly [3] or 70 million in the mouse [4]) and specializations with a great historical importance for neuroscience (e.g., “giant axons” [5] and “giant synapses” [6–8]).”

(2) L51, 279: "Octopodiformes" is a superorder, not a genus or a species name. It should not go in italics.

We updated this throughout the text.

(3) L53: "even smaller" seems odd here, because the argument of the sentence is to stress the large genome size of Octopodiformes. Perhaps start the sentence by stating that it is sometimes smaller, but often larger.

We rephrased the sentence for clarity, it now reads (lines 55-58):

“While the genomes of Octopodiformes (Octopus, Eledone, Argonauta) are either smaller than (1.1 Gigabases or Gb [45]) or comparable in size to that of humans (around 3 Gb [46,47]) the typical genomes of Decapodiformes (squids and cuttlefish) often reach 6 Gb [48,49].”

(4) L90: What tool was used to estimate the k-mer distribution of the long reads? Jellyfish? FastK? It's not mentioned anywhere in the text.(5) L95: What k-mer size did the authors use to estimate k-mer distribution?

We thank the reviewer for pointing out this missing information, and have included the details in the methods (lines 692-694):

“The k-mer distribution was estimated using Meryl [165] within the Merfin [166] package with a k-mer size of 21, and genomeGenome size was estimated using GenomeScope [77] from Illumina short reads and PacBio HiFi data.”

(6) L99: What about using the most recent BUSCO databases? odb12?

We thank the reviewer for this question, which prompted us to compute BUSCO scores using the more recent *odb12* database. The results are shown in Supplementary Figure 2C. Both gene sets have been refined by including more species and using a more stringent filtering approach, so the more recent database contains fewer and more conserved genes [4]. For the *mollusca* gene sets, a great improvement in completeness was observed between *odb10* and *odb12* (Supplementary Figure 2C); the *metazoan* completeness was marginally increased. Therefore, we evaluated all new assemblies produced since the first submission with the *odb12* database.

(7) L107: How many scaffolds were obtained in total? After manual curation, how many of the scaffolds were placed in the "correct" chromosomes? How many scaffolds were in the shrapnel? Were these scaffolds mostly repetitive regions? Or did they contain important genetic information?

These are important questions. To evaluate the content of the “shrapnel”, we split the manually curated assembly into the 47 chromosomes and the 1840 residual scaffolds, and computed BUSCO scores for both. While the 47 chromosome scaffolds contain the majority of conserved genes: C:92.9%[S:92.7%,D:0.1%],F:4.0%,M:3.1% with *metazoa_odb12* and C:88.7%[S:88.0%,D:0.7%],F:4.4%,M:6.9% with *mollusca_odb12*, the unplaced scaffolds still contain a few BUSCOs: C:2.5%[S:2.4%,D:0.1%],F:2.4%,M:95.1% from *metazoa_odb12* and C:1.9%[S:1.7%,D:0.2%],F:1.2%,M:96.9% from *mollusca_odb12*. Even if only a few BUSCOs are present on these scaffolds, it means they contain important genetic information. Additionally, we observed low, but non-zero alignment of RNA reads to these scaffolds. We observed a slightly elevated repeat content in the unplaced scaffolds (Author response image 2), and a variable base composition (Figure 1C) compared to the chromosome scaffolds.

**Author response image 2. sa4fig2:** Quantification of repeat content in chromosome scaffolds and unplaced residual scaffolds. Density plot showing fraction of repeat masked bases in total sequence length for chromosome scaffolds (i.e. scaffolds 1-47) in teal and all remaining small scaffolds (1840 scaffolds) in purple. Median repeat fraction is shown as vertical lines.

The slightly elevated repeat content in the unplaced scaffolds provides a likely explanation for their fragmented state: repeat-rich regions are inherently difficult to assemble and scaffold, as repetitive sequences cause ambiguous read alignments that prevent contigs from being confidently joined or anchored to chromosomal scaffolds during HiC-based scaffolding. This is consistent with the near-complete absence of BUSCO genes from the unplaced scaffolds - not because these fragments lack biologically relevant sequence entirely, as evidenced by the residual BUSCO hits and RNA read alignments, but because the gene-rich portions of the genome are largely captured in the 47 chromosome scaffolds. The unplaced scaffolds instead likely represent fragmented contigs from repetitive or low-complexity genomic regions, such as centromeres, telomeres, and transposable element clusters, where assembly graph complexity and collapsed repeats prevent confident placement. The variable base composition further supports this interpretation, as GC-extreme or low-complexity sequences are disproportionately represented in assembly shrapnel. Together, these observations suggest that the unplaced scaffolds contain limited unique coding content but reflect genuine repeat-rich genomic sequence that cannot currently be placed without additional long-range information, such as optical mapping or ultra-long reads.

(8) L33, 53, 240, 255, 279: Decapodiformes, not in italics.

We changed this throughout the text.

(9) L228: Can you put this expansion in perspective with other taxa?

We added a more detailed comparison of our gene family expansion with different species to the revised manuscript, as detailed in response 4 to reviewer 1.

(10) L251: "However, our results show how difficult it still is to assemble large genomes with high karyotype numbers." Can you clarify how your results show this, because it is equally spectacular to assemble the karyotype with only PacBio and Hi-C data (and no linkage mapping).

Indeed, it is correct that the recent improvements in data quality and scaffolding algorithms enable these “spectacular” chromosome-scale assemblies without the need for linkage mapping. This sentence reflected our expectation to resolve a clear karyotype as has been demonstrated for multiple cephalopod genomes in recent years, including two cuttlefish species (*Octopus bimaculoides*, *Octopus vulgaris*, *Euprymna scolopes*, *Euprymna berryi*, *Acanthosepion lycidas* and *Acanthosepion esculenta*). To our knowledge, none of these publications used linkage mapping or cytogenetic methods to confirm the karyotype. In this light, our resulting chromosome number and the discrepancy to a second assembly of the same species led us to this conclusion. We updated the section in the revised discussion as follows (lines 466-473):

“Taken together, our results illustrate the difficulty of assembling large genomes with high repeat content and large karyotypes, at least from sequencing data alone. Internal validation methods and genome comparisons across species are therefore important. Convergence of reliable estimates will, in turn, help identify chromosomal fusion-with-mixing events (FWM; fusion of two ancestral chromosomes followed by extensive shuffling of their gene content) that are clade specific. Early branching order in Decapodiformes has been notoriously unstable [53,84,94,144–147]; thus, such rare and irreversible FWM characters could be useful in further phylogenetic analysis of this clade [51,148].”

(11) L419: Why use the phased haplotype 1 instead of the primary assembly generated by hifiasm?

We thank the reviewer for this important question. We used the phased haplotype assembly because it provides a biologically coherent representation with the least amount of duplication by avoiding allele-collapsing and haplotype-switching that can be present in the primary assembly. We reasoned that this would result in clearer gene models and a more accurate representation of structural variation. However, we acknowledge that this comes at the cost of reduced contiguity and completeness, as becomes apparent in our BUSCO comparison shown in Supplementary Figure 2, where the phased haplotypes have fewer duplicated genes than the primary assembly, but more missing genes in turn. When reassembling both datasets for our comparison, we used the primary assembly to use the longest contigs as input for scaffolding.

(12) L444: It is unclear from what tissues and life stages RNA-seq data were used or were available from other species.

This is an important detail. RNA-seq data was collected from two adult *Sepia officinalis*, from various tissues (whole brain, retina, skin, mantle, arm, tentacle). For the long-read PacBio Isoseq data, tissue was taken from the animal used for genome sequencing (6 months old), and tissue for short-read Illumina RNA-seq was taken from another adult (8 months old). The data have been released on SRA (study accession SRP570862), where all sample details are listed as well. We added the SRA accession to the data availability section of the revised manuscript. We clarified the relevant sections in the methods:

lines 628-629:

“RNA was isolated from various flash-frozen tissues (different brain areas, mantle/epidermis, arm/tentacle; 5-10 mg each).”

lines 678-680:

“For short-read RNA sequencing, tissue from another animal (8-month-old adult, F0 from eggs collected in Normandie, France) was used. RNA was isolated from various flash-frozen tissues (different brain areas, skin and retina; 5 mg each).”

(13) L454, 469: Why is minimap2 in italics? It wasn't formatted like this before. Same for StringTie.

We thank the reviewer for their detailed methods review. In the updated methods section, all formatting of used softwares was harmonized.

(14) L461: Lophotrochozoa is a clade, not a genus or species. Not in italics.

This is now changed throughout the revised manuscript.

(15) Figure 1D: Axes labels are hard to read.

We have now increased the axis label size.

(16) Figure 2: Consider increasing font sizes. Many chromosome orientations seem to be flipped across species, which makes it harder to see smaller-scale rearrangements or notice less conserved chromosomes. Would it make sense to standardize these?

We increased the font sizes and plotted only fully collinear syntenic blocks (instead of aggregated syntenic regions, the default of GENESPACE) for improved readability.

References:

Below are references cited in our responses. References from the reproduced manuscript sections are included in the revised manuscript.

(1) Secomandi, S., Gallo, G.R., Rossi, R., Rodríguez Fernandes, C., Jarvis, E.D., Bonisoli-Alquati, A., Gianfranceschi, L., and Formenti, G. (2025). Pangenome graphs and their applications in biodiversity genomics. Nat. Genet. 57, 13–26. https://doi.org/10.1038/s41588-024-02029-6.

(2) Open2C, Abdennur, N., Fudenberg, G., Flyamer, I.M., Galitsyna, A.A., Goloborodko, A., Imakaev, M., and Venev, S.V. (2023). Pairtools: from sequencing data to chromosome contacts. Preprint at bioRxiv, https://doi.org/10.1101/2023.02.13.528389
https://doi.org/10.1101/2023.02.13.528389.

(3) Lupše, N., Reid, A., Taite, M., Kubodera, T., and Allcock, A.L. (2023). Cuttlefishes (Cephalopoda, Sepiidae): the bare bones—an hypothesis of relationships. Mar. Biol. 170, 93. https://doi.org/10.1007/s00227-023-04195-3.

(4) Tegenfeldt, F., Kuznetsov, D., Manni, M., Berkeley, M., Zdobnov, E.M., and Kriventseva, E.V. (2025). OrthoDB and BUSCO update: annotation of orthologs with wider sampling of genomes. Nucleic Acids Res. 53, D516–D522. https://doi.org/10.1093/nar/gkae987.